# Constrained Cross-Entropy Method for Safe Reinforcement Learning

**Min Wen**

Department of Electrical and Systems Engineering
University of Pennsylvania
wenm@seas.upenn.edu

**Ufuk Topcu**

Department of Aerospace Engineering and Engineering Mechanics
University of Texas at Austin
utopcu@utexas.edu

## Abstract

We study a safe reinforcement learning problem in which the constraints are defined as the expected cost over finite-length trajectories. We propose a constrained cross-entropy-based method to solve this problem. The method explicitly tracks its performance with respect to constraint satisfaction and thus is well-suited for safety-critical applications. We show that the asymptotic behavior of the proposed algorithm can be almost-surely described by that of an ordinary differential equation. Then we give sufficient conditions on the properties of this differential equation for the convergence of the proposed algorithm. At last, we show with simulation experiments that the proposed algorithm can effectively learn feasible policies without assumptions on the feasibility of initial policies, even with non-Markovian objective functions and constraint functions.

## 1 Introduction

This paper studies the following constrained optimal control problem: given a dynamical system model with continuous states and actions, a objective function and a constraint function, find a controller that maximizes the objective function while satisfying the constraint. Although this topic has been studied for decades within the control community [3], it is still challenging for practical problems. To illustrate some major difficulties, consider the synthesis of a policy for a nonholonomic mobile robot to reach a goal while avoiding obstacles (which introduces constraints) in a cost-efficient way (which induces an objective). The obstacle-free state space is usually nonconvex. The equations of the dynamical system model are typically highly nonlinear. Constraint functions and cost functions may not be convex or differentiable in the state and action variables. There may even be hidden variables that are not observable and make transitions and costs non-Markovian. Given all these difficulties, we still need to compute a policy that is at least feasible and improve the cost objective as much as possible.

Reinforcement learning (RL) methods have been widely used to learn optimal policies for agents with complicated or even unknown dynamics. For problems with continuous state and action spaces, the agent's policy is usually modeled as a parameterized function of states such as deep neural networks and later trained using policy gradient methods [35; 30; 27; 28; 21; 8; 29]. By encoding control tasks as reward or cost functions, RL has successfully solved a wide range of tasks such as Atari games [19; 20], the game of Go [31; 32], controlling simulated robots [36; 24] and real robots [15; 37; 22].

Most of the existing methods for RL solve only unconstrained problems. However, it is generally non-trivial to transform a constrained optimal control problem into an unconstrained one, due to the asymmetry between the goals of objective optimization and constraint satisfaction. On the one hand, it is usually acceptable to output a policy that is only locally optimal with respect to the optimization objective. On the other hand, in many application scenarios where constraints encode safety requirements or the amount of available resources, violating the constraint even by a small amount may have significant consequences.

Existing methods for safe reinforcement learning that are based on policy gradient methods cannot guarantee strict feasibility of the policies they output, even when initialized with feasible initial policies. When initialized with an infeasible policy, they usually are not be able to find even a single feasible policy until their convergence (with an example in Section 5). These limitations motivate the following question: Can we develop a reinforcement learning algorithm that explicitly addresses the priority of constraint satisfaction? Rather than assuming that the initial policy is feasible and that one can always find a feasible policy in the estimated gradient direction, we need to deal with cases in which the initial policy is not feasible, or we have never seen a feasible policy before.

Inspired by stochastic optimization methods based on the cross-entropy (CE) concept [11], we propose a new safe reinforcement learning algorithm, which we call the *constrained cross-entropy (CCE)* method. The basic framework is the same with standard CE methods: In each iteration, we sample from a distribution of policies, select a set of elite sample policies and use them to update the policy distribution. Rather than treating the constraints as an extra term in the objective function as what policy gradient method do, we use constraint values to sort sample policies. If there are not enough feasible sample policies, we select only those with the best constraint performance as elite sample policies. If a given proportion of the sample policies are feasible, we select the feasible sample policies with the best objective values as elite sample policies. Instead of initializing the optimization with a feasible policy, the method improves both the objective function and the constraint function with the constraint as a prioritized concern.

Our algorithm can be used as a black-box optimizer. It does not even assume that there is an underlying reward or cost function encoding the optimization objective and constraint functions. In fact, the algorithm can be applied to any finite-horizon problem (say, with horizon $N$) whose objective and constraint functions are defined as the average performance over some distribution of trajectories. For example, a constraint function can be the probability that the agent satisfies a given task specification (which may be Markovian or non-Markovian) with policy $\pi_\theta$, if the satisfaction of the given task can be decided with any $N$-step trajectory. An optimization objective may be the expected number of steps before the agent reaches a goal state, or the expected maximum distance the agent has left from its origin, or the expected minimum distance between the agent and any obstacle over the whole trajectory.

Our contributions are as follows. First, we present a model-free constrained RL algorithm that works with continuous state and action spaces. Second, we prove that the asymptotic behavior of our algorithm can be almost-surely described by that of an ordinary differential equation (ODE), which is easily interpretable with respect to the objectives. Third, we give sufficient conditions on the properties of this ODE to guarantee the convergence of our algorithm. We show with numerical experiments that, our algorithm converges to feasible policies in all our experiments with all combinations of feasible or infeasible initial policies, Markovian or non-Markovian objectives and constraints, while other policy-gradient-based algorithms fail to find strictly feasible solutions.

## 2   Related Work

Safety has long been concerned in RL literature and is formulated as various criteria [7]. We choose to take the so-called *constrained criterion* [7] to encode our safety requirement, which is the same as in the literature of constrained Markov decision processes (CMDP) [2]. Approaches are still limited for safe RL with continuous state and action spaces. Uchibe and Doya [34] proposed a constrained policy gradient reinforcement learning algorithm, which relies on projected gradients to maintain feasibility. The computation of projection restricts the types of constraints it can deal with, and there is no known guarantee on convergence. Chow et al. [4] came up with a trajectory-based primal-dual subgradient algorithm for a risk-constrained RL problem with finite state and action spaces. The algorithm is proved to converge almost-surely to a local saddle point. However, the constraints are

just implicitly considered by updating dual variables and the output policy may not actually satisfy the constraints. Recently, Achiam et al. [1] proposed a trust region method for CMDP called constrained policy optimization (CPO), which can deal with high-dimensional policy classes such as neural networks and claim to maintain feasibility if started with a feasible solution. However, we found in Section 5 that feasibility is rarely guaranteed during learning in practice, possibly due to errors in gradient and Hessian matrix estimation.

Cross-entropy-based stochastic optimization techniques have been applied to a series of RL and optimal control problems. Mannor et al. [18] used cross-entropy methods to solve a stochastic shortest-path problem on finite Markov decision processes, which is essentially an unconstrained problem. Szita and Lőrincz [33] took a noisy variant to learn how to play Tetris. Kobilarov [14] introduced a similar technique to motion planning in constrained continuous-state environments by considering distributions over collision-free trajectories. Livingston et al. [17] generalized this method to deal with a broader class of trajectory-based constraints called linear temporal logic specifications. Both methods simply discard all sample trajectories that violate the given constraints, and thus their work can be considered as a special case of our work when the constraint function has binary outputs. Similar applications in approximate optimal control with constraints can be found in [23; 6; 16].

## 3 Preliminaries

For a set $B$, let $\mathcal{D}(B)$ be the set of all probability distributions over $B$, $int(B)$ be the interior of $B$ and $B^k := \{s_0, s_1, \ldots, s_{k-1} \mid s_t \in B, \ \forall t = 0, \ldots, k-1\}$ be the set of all sequences composed by elements in $B$ of length $k$ for any $k \in \mathbb{N}^+$.

A (reward-free) *Markov decision process (MDP)* is defined as a tuple $(\mathcal{S}, \mathcal{A}, T, P_0)$, where $\mathcal{S}$ is a set of states, $\mathcal{A}$ is a set of actions, $T : \mathcal{S} \times \mathcal{A} \to \mathcal{D}(\mathcal{S})$ is a transition distribution function and $P_0 \in \mathcal{D}(\mathcal{S})$ is an initial state distribution. Let $\Pi : \mathcal{S} \to \mathcal{D}(\mathcal{A})$ be the set of all stationary policies. Given a finite horizon $N$, an $N$-*step trajectory* is a sequence of $N$ state-action pairs. Each stationary policy $\pi \in \Pi$ decides a distribution over $N$-step trajectories such that the probability to draw a trajectory $\tau = s_0, a_0, \ldots, s_{N-1}, a_{N-1}$ is $P_{\pi,N}(\tau) = P_0(s_0) \prod_{t=0}^{N-2} T(s_{t+1}|s_t, a_t) \prod_{t=0}^{N-1} \pi(a_t|s_t)$. Without loss of generality, we assume that $N$ is fixed and use $P_\pi$ to represent $P_{\pi,N}$.

An *objective* function $J : (\mathcal{S} \times \mathcal{A})^N \to \mathbb{R}$ is a mapping from each $N$-step trajectory to a scalar value. For each $\pi \in \Pi$, let

$$G_J(\pi) := \mathbb{E}_{\tau \sim P_\pi}[J(\tau)]$$

be the expected value of $J$ with the $N$-step trajectory distribution decided by $\pi$. A policy $\pi \in \Pi$ is an *optimal* policy in $\Pi$ with respect to $J$ if $G_J(\pi) = \max_{\pi' \in \Pi} G_J(\pi')$.

A *cost* function $Z : (\mathcal{S} \times \mathcal{A})^N \to \mathbb{R}$ is also a function defined on $N$-step trajectories. Let

$$H_Z(\pi) := \mathbb{E}_{\tau \sim P_\pi}[Z(\tau)]$$

be the expected cost over trajectory distribution $P_\pi$. A policy $\pi \in \Pi$ is *feasible* for a constrained optimization problem with cost function $Z$ and *constraint upper bound* $d$ if $H_Z(\pi) \le d$. Let $\Pi_{Z,d}$ be the set of all feasible policies.

For notational simplicity, we omit $J$ and $Z$ in $G_J$ and $H_Z$ whenever there is no ambiguity. For any policy $\pi \in \Pi$, we refer to $G(\pi)$ and $H(\pi)$ as the $G$-*value* and $H$-*value* of $\pi$.

## 4 Constrained Cross-Entropy Framework

### 4.1 Problem Formulation

In this paper, we consider a finite-horizon RL problem with a strictly positive objective function $J : (\mathcal{S} \times \mathcal{A})^N \to \mathbb{R}^+$, a cost function $Z : (\mathcal{S} \times \mathcal{A})^N \to \mathbb{R}$ and a constraint upper bound $d$. For MDPs with continuous state and action spaces, it is usually intractable to exactly solve an optimal stationary policy due to the curse of dimensionality. An alternative is to use function approximators, such as neural networks, to parameterize a subset of policies. Given a parameterized class of policies $\Pi_\Theta$ with a parameter space $\Theta \subseteq \mathbb{R}^{d_\theta}$, we aim to solve the following problem:

$$\pi^* = \underset{\pi \in \Pi_\Theta \bigcap \Pi_{Z,d}}{\arg\max} \ G_J(\pi).$$

The proposed algorithm, which we call the *constrained cross-entropy* method, generalizes the well-known cross-entropy method [18] for unconstrained optimization. The basic idea is to generate a sequence of policy distributions that eventually concentrates on a feasible (locally) optimal policy. Given a distribution over $\Pi_\Theta$, we randomly generate a set of sample policies, sort them with a ranking function that depends on their $G$-values and $H$-values and then update the policy distribution with a subset of high-ranking sample policies.

Given the policy parameterization $\Pi_\Theta$, we use distributions over the parameter space $\Theta$ to represent distributions over the policy space $\Pi_\Theta$. We focus ourselves on a specific family of distributions over $\Theta$ called *natural exponential family* (NEF), which includes many useful distributions such as Gaussian distribution and Gamma distribution. A formal definition of NEF is as follows.

**Definition 4.1.** A parameterized family $F_\mathcal{V} = \{f_{\boldsymbol{v}} \in \mathcal{D}(\Theta), \boldsymbol{v} \in \mathcal{V} \subseteq \mathbb{R}^{d_v}\}$ is called a *natural exponential family* if there exist continuous mappings $\Gamma : \mathbb{R}^{d_\theta} \to \mathbb{R}^{d_v}$ and $K : \mathbb{R}^{d_\theta} \to \mathbb{R}$ such that $f_{\boldsymbol{v}}(\theta) = \exp\left(\boldsymbol{v}^\mathsf{T}\Gamma(\theta) - K(\boldsymbol{v})\right)$, where $\mathcal{V} \subseteq \{\boldsymbol{v} \in \mathbb{R}^{d_v} : |K(\boldsymbol{v})| < \infty\}$ is the natural parameter space and $K(\boldsymbol{v}) = \log \int_\Theta \exp\left(\boldsymbol{v}^\mathsf{T}\Gamma(\theta)\right) d\theta$.

As with other CE-based algorithms, we replace the original objective $G(\pi_\theta) = \mathbb{E}_{\tau \sim P_{\pi_\theta}}[J(\tau)]$ with a surrogate function. For the unconstrained CE method, the surrogate function is the conditional expectation over policies whose $G$-values are highly ranked with the current sampling distribution $f_{\boldsymbol{v}}$. The ranking function is defined using the concept of $\rho$-quantiles for random variables, which is formally defined as below.

**Definition 4.2.** [10] Given a distribution $P \in \mathcal{D}(\mathbb{R})$, $\rho \in (0,1)$ and a random variable $X \sim P$, the $\rho$-quantile of $X$ is defined as a scalar $\gamma$ such that $Pr(X \leq \gamma) \geq \rho$ and $Pr(X \geq \gamma) \geq 1 - \rho$.

For $\rho \in (0,1)$, $\boldsymbol{v} \in \mathcal{V}$ and any function $X : \Theta \to \mathbb{R}$, we denote the $\rho$-quantile of $X$ for $\theta \sim f_{\boldsymbol{v}}$ by $\xi_X(\rho, \boldsymbol{v})$. We also define $\delta : \mathbb{R} \times \{\geq, \leq, >, <, =\} \times \mathbb{R} \to \{0,1\}$ as an indicator function such that for $\circ \in \{\geq, \leq, >, <, =\}$, $\delta(x \circ y) = 1$ if and only if $x \circ y$ holds. The surrogate objective function for the unconstrained CE method is $\mathbb{E}_{\theta \sim f_{\boldsymbol{v}}}[G(\pi_\theta)\delta(G(\pi_\theta) \geq \xi_G(1 - \rho, \boldsymbol{v}))]$. In other words, a policy $\pi_\theta$ is considered as highly ranked if $G(\pi_\theta) \geq \xi_G(1 - \rho, \boldsymbol{v})$. When there is a constraint $H(\pi) \leq d$, we define $U : \Pi_\Theta \to \mathbb{R}$ such that $U(\pi_\theta) := G(\pi_\theta)\delta(H(\pi_\theta) \leq d)$ for any $\theta \in \Theta$ and extend the surrogate function as follows:

$$L(\boldsymbol{v}; \rho) := \begin{cases} \mathbb{E}_{\theta \sim f_{\boldsymbol{v}}}[G(\pi_\theta)\delta(H(\pi_\theta) \leq \xi_H(\rho, \boldsymbol{v}))], & \text{if } \xi_H(\rho, \boldsymbol{v}) > d; \\ \mathbb{E}_{\theta \sim f_{\boldsymbol{v}}}[U(\pi_\theta)\delta(U(\pi_\theta) \geq \xi_U(1 - \rho, \boldsymbol{v}))], & \text{otherwise.} \end{cases} \quad (1)$$

We can combine the two cases. Define $S : \Pi_\Theta \times \mathcal{V} \times (0,1) \to \{0,1\}$ such that

$$\begin{aligned} S(\pi_\theta, \boldsymbol{v}, \rho) :=& \delta(\xi_H(\rho, \boldsymbol{v}) > d)\delta(H(\pi_\theta) \leq \xi_H(\rho, \boldsymbol{v})) + \\ & \delta(\xi_H(\rho, \boldsymbol{v}) \leq d)\delta(H(\pi_\theta) \leq d)\delta(U(\pi_\theta) \geq \xi_U(1 - \rho, \boldsymbol{v})), \end{aligned}$$

then (1) can be rewritten as

$$L(\boldsymbol{v}; \rho) = \mathbb{E}_{\theta \sim f_{\boldsymbol{v}}}[G(\pi_\theta)S(\pi_\theta, \boldsymbol{v}, \rho)]. \quad (2)$$

The interpretation of $L$ is as follows: If the $\rho$-quantile of $H$ for the current policy distribution $f_{\boldsymbol{v}}$ is greater than the constraint threshold $d$, we select policies in $\Pi_\Theta$ by their $H$-values in order to increase the probability of drawing feasible policies. Consequently, $\pi_\theta$ is highly ranked if $H(\pi_\theta) \leq \xi_H(\rho, \boldsymbol{v})$. If the proportion of feasible policies is higher than $\rho$, we select policies that are both feasible and with large objective values, i.e., $\pi_\theta$ is highly ranked if $H(\pi_\theta) \leq d$ and $U(\pi_\theta) \geq \xi_U(1 - \rho, \boldsymbol{v})$. Intuitively, $S$ can be considered as the indicator function of the highly-ranked or elite samples.

*Remark* 1. By maximizing $U$, we implicitly prioritizes feasibility over the $G$ objective: For any feasible policy $\pi$ and infeasible policy $\pi'$, it can be easily verified that $U(\pi) \geq U(\pi')$, as $G$ and $U$ are non-negative by definition.

*Remark* 2. If $\xi_H(\rho, \boldsymbol{v}) \leq d$, then $G(\pi_\theta)\delta(G(\pi_\theta) \geq \xi_G(1 - \rho, \boldsymbol{v})) \geq U(\pi_\theta)\delta(U(\pi_\theta) \geq \xi_U(1 - \rho, \boldsymbol{v})) \geq G(\pi_\theta)\delta(H(\pi_\theta) \leq \xi_H(\rho, \boldsymbol{v}))$. Intuitively, if at least $100\rho\%$ of all policies are feasible, $L(\boldsymbol{v}; \rho)$ is less than the objective value for the unconstrained CE method and greater than the expected $G$-value over the $100\rho\%$ policies of the highest $H$-values.

The main problem we solve in this paper can be then stated as follows.

---

**Algorithm 1** Constrained Cross-Entropy Method

---

**Require:** An objective function $G$, a constraint function $H$, a constraint upper bound $d$, a class of parameterized policies $\Pi_\Theta$, an NEF family $F_\mathcal{V}$.

1: $l \leftarrow 1$. Initialize $n_l, \boldsymbol{v}_l, \rho, \lambda_l, \alpha_l$. $k_l \leftarrow \lceil \rho n_l \rceil$. $\hat{\eta}_l \leftarrow \mathbf{0}$.
2: **repeat**
3:     Sample $\theta_1, \ldots, \theta_{n_l} \sim f_{\boldsymbol{v}_l}$ i.i.d..
4:     **for** $i = 1, \ldots, n_l$ **do**
5:         Simulate $\pi_{\theta_i}$ and estimate $G(\pi_{\theta_i}), H(\pi_{\theta_i})$.
6:     **end for**
7:     Sort $\{\theta_i\}_{i=1}^{n_l}$ in ascending order of $H$. Let $\Lambda_l$ be the first $k_l$ elements.
8:     **if** $H(\pi_{\theta_{k_l}}) \leq d$ **then**
9:         Sort $\{\theta_i \mid H(\pi_{\theta_i}) \geq d\}$ in descending order of $G$. Let $\Lambda_l$ be the first $k_l$ elements.
10:    **end if**
11:    $\hat{\eta}_{l+1} \leftarrow \alpha_l \sum_{\theta \in \Lambda_l} \frac{G(\pi_\theta)}{\sum_{\theta \in \Lambda_l} G(\pi_\theta)} \Gamma(\theta) + (1 - \alpha_l)\left(\frac{\lambda_l}{n_l} \sum_{i=1}^{n_l} \Gamma(\theta_i) + (1 - \lambda_l)\hat{\eta}_l\right)$.
12:    $\boldsymbol{v}_{l+1} \leftarrow m^{-1}(\hat{\eta}_{l+1})$.
13:    Update $n_l, \lambda_l, \alpha_l$. $l \leftarrow l + 1$. $k_l \leftarrow \lceil \rho n_l \rceil$.
14: **until** Stopping rule is satisfied.

---

**Problem 1.** *Given a set $\Pi = \{\pi_\theta : \theta \in \Theta\}$ of policies with parameter space $\Theta$, an NEF $F_\mathcal{V} = \{f_{\boldsymbol{v}} \in \mathcal{D}(\Theta) : \boldsymbol{v} \in \mathcal{V}\}$ of distributions over $\Theta$, two functions $G : \Pi \to \mathbb{R}^+$ and $H : \Pi \to \mathbb{R}$, a constraint upper bound $d$ and $\rho \in (0, 1)$, compute $\boldsymbol{v}^* \in \mathcal{V}$ such that*

$$\boldsymbol{v}^* = \arg\max_{\boldsymbol{v} \in \mathcal{V}} L(\boldsymbol{v}; \rho),$$

*where $L : \mathcal{V} \times (0, 1) \to \mathbb{R}$ is defined in* (1) *or* (2).

## 4.2 The Constrained Cross-Entropy Algorithm

The pseudocode of the constrained cross-entropy algorithm is given in Algorithm 1. We first explain the basic ideas behind the updates in Algorithm 1, and provide a proof of convergence in Section 4.3.

We first describe the key idea behind the (idealized) CE-based stochastic optimization method as in [12]. For notational simplicity, we use $\mathbb{E}_{\boldsymbol{v}}[\cdot]$ to represent $\mathbb{E}_{\theta \sim f_{\boldsymbol{v}}}[\cdot]$ in the rest of this paper. Define $m(\boldsymbol{v}) := \mathbb{E}_{\boldsymbol{v}}[\Gamma(\theta)] \in \mathbb{R}^{d_v}$ for $\boldsymbol{v} \in \mathcal{V}$, which is continuously differentiable in $\boldsymbol{v}$ and $\frac{\partial}{\partial \boldsymbol{v}} m(\boldsymbol{v}) = \text{Cov}_{\boldsymbol{v}}[\Gamma(\theta)]$ where $\text{Cov}_{\boldsymbol{v}}[\Gamma(\theta)]$ denotes the covariance matrix of $\Gamma(\theta)$ with $\theta \sim f_{\boldsymbol{v}}$. We take Assumption 1 to guarantee that $m^{-1}$ exists and is continuously differentiable over $\{\eta : \exists \boldsymbol{v} \in int(\mathcal{V}) \; s.t. \; \eta = m(\boldsymbol{v})\}$ (see Lemma 1 in supplemental material).

**Assumption 1.** $\text{Cov}_{\boldsymbol{v}}[\Gamma(\theta)]$ *is positive definite for any $\boldsymbol{v} \in \mathcal{V} \subseteq int(\{\boldsymbol{v} \in \mathbb{R}^{d_v} : |K(\boldsymbol{v})| < \infty\})$.*

By definition of $\rho$-quantiles, it is a rare event to sample the highly ranked policies for small $\rho$. Thus we apply importance sampling to estimate $L(\boldsymbol{v}; \rho)$ using any sampling distribution $g$ that shares the same support $\Theta$ as $f_{\boldsymbol{v}}$, among which the optimal distribution $g_{\boldsymbol{v}}^*$ [25] with minimal variance is

$$g_{\boldsymbol{v}}^*(\theta) = \frac{G(\pi_\theta)S(\pi_\theta, \boldsymbol{v}, \rho)f_{\boldsymbol{v}}(\theta)}{L(\boldsymbol{v}; \rho)}. \tag{3}$$

In practice we smoothen the updates by including a learning rate $\alpha \in (0, 1)$ so the goal distribution is $\tilde{g}_{\boldsymbol{v}} = \alpha g_{\boldsymbol{v}}^* + (1 - \alpha)f_{\boldsymbol{v}}$. We can project $\tilde{g}_{\boldsymbol{v}}$ to $f_{\boldsymbol{v}'} \in F_\mathcal{V}$ by minimizing the Kullback-Leibler (KL) divergence of $f_{\boldsymbol{v}''} \in F_\mathcal{V}$ from $\tilde{g}_{\boldsymbol{v}}$, which is equivalent to minimizing the cross entropy between $\tilde{g}_{\boldsymbol{v}}$ and $f_{\boldsymbol{v}''}$. If $F_\mathcal{V}$ is an NEF, $\log f_{\boldsymbol{v}''}(\theta) = (\boldsymbol{v}'')^\intercal \Gamma(\theta) - K(\boldsymbol{v}'')$ is concave in $\boldsymbol{v}''$; thus $-\int_\Theta \tilde{g}_{\boldsymbol{v}}(\theta) \log f_{\boldsymbol{v}''}(\theta)d\theta$ is convex in $\boldsymbol{v}''$. As a result, $\boldsymbol{v}'$ can be found by setting $\frac{\partial}{\partial \boldsymbol{v}''}\left(-\int_\Theta \tilde{g}_{\boldsymbol{v}}(\theta) \log f_{\boldsymbol{v}''}(\theta)d\theta\right) = \mathbf{0}$, which induces

$$m(\boldsymbol{v}') - m(\boldsymbol{v}) = \alpha\left(\mathbb{E}_{g_{\boldsymbol{v}}^*}[\Gamma(\theta)] - m(\boldsymbol{v})\right). \tag{4}$$

As a property of NEF, the KL-divergence of $f_{\boldsymbol{v}}$ from $g$ satisfies $\frac{\partial}{\partial \boldsymbol{v}} D_{KL}(g, f_{\boldsymbol{v}}) = -\mathbb{E}_g[\Gamma(\theta)] + m(\boldsymbol{v})$. Therefore

$$m(\boldsymbol{v}') - m(\boldsymbol{v}) = -\alpha\left(\frac{\partial}{\partial \boldsymbol{v}''} D_{KL}(g_{\boldsymbol{v}}^*, f_{\boldsymbol{v}''})\right)\Big|_{\boldsymbol{v}''=\boldsymbol{v}}, \tag{5}$$

which confirms that $m(\boldsymbol{v})$ is always updated in the negative gradient direction of the objective function $D_{KL}(g_{\boldsymbol{v}}^*, f_{\boldsymbol{v}})$ where $g_{\boldsymbol{v}}^*$ is the optimal sampling distribution from importance sampling.

*Remark* 3. The equality in (5) holds not just for the optimal distribution $g_{\boldsymbol{v}}^*$ but for any reference distribution.

Define $\tilde{L}(\boldsymbol{v}; \rho) := \mathbb{E}_{g_{\boldsymbol{v}}^*}[\Gamma(\theta)] - m(\boldsymbol{v})$. If $G$ has a strictly positive lower bound and is bounded, then

$$
\begin{aligned}
\tilde{L}(\boldsymbol{v}; \rho) &= \frac{\mathbb{E}_{\boldsymbol{v}}[G(\pi_\theta)S(\pi_\theta, \boldsymbol{v}, \rho)\Gamma(\theta)]}{L(\boldsymbol{v}; \rho)} - m(\boldsymbol{v}) \\
&= \int_\Theta \frac{G(\pi_\theta)S(\pi_\theta, \boldsymbol{v}, \rho)}{L(\boldsymbol{v}; \rho)} f_{\boldsymbol{v}}(\theta)(\Gamma(\theta) - m(\boldsymbol{v}))d\theta \\
&\overset{(*)}{=} \int_\Theta \frac{G(\pi_\theta)S(\pi_\theta, \boldsymbol{v}, \rho)}{L(\boldsymbol{v}; \rho)} \Big(\frac{\partial}{\partial \boldsymbol{v}} f_{\boldsymbol{v}}(\theta)\Big)d\theta \overset{(**)}{=} \frac{\partial}{\partial \boldsymbol{v}''} \frac{\mathbb{E}_{\boldsymbol{v}''}[G(\pi_\theta)S(\pi_\theta, \boldsymbol{v}, \rho)]}{L(\boldsymbol{v}; \rho)}\Big|_{\boldsymbol{v}''=\boldsymbol{v}} \\
&= \frac{\partial}{\partial \boldsymbol{v}''} \log \mathbb{E}_{\boldsymbol{v}''}[G(\pi_\theta)S(\pi_\theta, \boldsymbol{v}, \rho)]\Big|_{\boldsymbol{v}''=\boldsymbol{v}},
\end{aligned}
\tag{6}
$$

where the $(*)$ step holds by noticing $\frac{\partial}{\partial \boldsymbol{v}} f_{\boldsymbol{v}}(\theta) = f_{\boldsymbol{v}}(\theta)(\Gamma(\theta) - m(\boldsymbol{v}))$ and the $(**)$ step holds by the dominated convergence theorem. Combining (4) and (6), we get

$$
m(\boldsymbol{v}') - m(\boldsymbol{v}) = \alpha \frac{\partial}{\partial \boldsymbol{v}''} \log \mathbb{E}_{\boldsymbol{v}''}[G(\pi_\theta)S(\pi_\theta, \boldsymbol{v}, \rho)]\Big|_{\boldsymbol{v}''=\boldsymbol{v}} = \alpha \tilde{L}(\boldsymbol{v}; \rho),
\tag{7}
$$

which leads to the second interpretation of the updates: The update from $\boldsymbol{v}$ to $\boldsymbol{v}'$ approximately follows the gradient direction of $\log L(\boldsymbol{v}''; \rho)$, while the quantiles are estimated using the previous distribution $f_{\boldsymbol{v}}$.

Algorithm 1 essentially takes the above updates in (3) and (4) in each iteration, with all expectations and quantiles estimated by Monte Carlo simulation. Given $f_{\boldsymbol{v}_l} \in \mathcal{D}(\Theta)$ in the $l^{th}$ iteration, we sample over policies (Step 3), evaluate their $G$-values and $H$-values (Step 5), estimate $S(\cdot, \boldsymbol{v}, \rho)$ (Step 7 to 10) and estimate $m(\boldsymbol{v}_{l+1})$ with $\hat{\eta}_{l+1}$ (Step 11) and finally update the sampling distribution to $\boldsymbol{v}_{l+1}$ (Step 12).

### 4.3 Convergence Analysis

We prove the convergence of Algorithm 1 by comparing the asymptotic behavior of $\{\hat{\eta}_l\}_{l \geq 0}$ with the flow induced by the following ordinary differential equation (ODE):

$$
\frac{\partial \eta(t)}{\partial t} = \tilde{L}(m^{-1}(\eta(t)); \rho),
\tag{8}
$$

where we define $\eta := m(\boldsymbol{v})$ or equivalently, $\boldsymbol{v} = m^{-1}(\eta)$.

We need a series of assumptions for technical reasons.

**Assumption 2.** *(2a) $\tilde{L}(\boldsymbol{v}; \rho)$ is continuous in $\boldsymbol{v} \in int(\mathcal{V})$ and (8) has a unique integral curve for any given initial condition.*

*(2b) The number of samples in the $l^{th}$ iteration is $n_l = \Theta(l^\beta)$, $\beta > 0$. The gain sequence $\{\alpha_l\}$ is positive and decreasing with $\lim_{l \to \infty} \alpha_l = 0$, $\sum_{l=1}^\infty \alpha_l = \infty$. $\{\lambda_l\}$ satisfies $\lambda_l = O(\frac{1}{l^\lambda})$ for some $\lambda > 0$ such that $\beta + 2\lambda > 1$.*

*(2c) For any $\rho \in (0, 1)$ and $f_{\boldsymbol{v}}$ for any $\boldsymbol{v} \in \mathcal{V}$, the $\rho$-quantile of $\{H(\pi_\theta) : \theta \sim f_{\boldsymbol{v}}\}$ and the $(1 - \rho)$-quantile of $\{U(\pi_\theta) : \theta \sim f_{\boldsymbol{v}}\}$ are both unique.*

*(2d) Both $\Theta$ and $\mathcal{V}$ are compact.*

*(2e) The function $G$ defined in Problem 1 is bounded and has a positive lower bound: $\inf_{\pi \in \Pi} G(\pi) > 0$. The function $H$ in Problem 1 is bounded.*

*(2f) $\boldsymbol{v}_l \in int(\mathcal{V})$ for any iteration $l$.*

Assumption (2a) ensures that (8) is well-posed and has a unique solution. Assumption (2b) addresses some requirements on the number of sampled policies in each iteration and other hyperparameters

in Algorithm 1. Assumptions (2c) to (2e) are used in the proof of the convergence of Algorithm 1. Assumption (2c) is required to show that $\frac{1}{n_l}\sum_{\theta \in \Lambda_l} G(\pi_\theta)$ in Step 11 of Algorithm 1 is an unbiased estimate of $\mathbb{E}_{\boldsymbol{v}_l}[G(\pi_\theta)S(\pi_\theta, \boldsymbol{v}_l, \rho)]$. Assumption (2d) and (2e) are compactness and boundedness constraints for the sets and functions involved in Algorithm 1, which are unlikely to be restrictive in practice. Assumption (2f) states that $\mathcal{V}$ is large enough such that the learned $\boldsymbol{v}$ lies within its interior.

The main result that connects the asymptotic behavior of Algorithm 1 with that of an ODE is stated in Theorem 4.1. The main idea behind the proof of Theorem 4.1 is similar to that of Theorem 3.1 in [12], although the details are tailored to our problem. There are two major parts in the convergence proof: The first part shows that all the sampling-based estimates converge to the true values almost surely, including sample quantiles and sample estimates of $G$, $H$ and $L$. The second part shows that the asymptotic behavior of the idealized updates in (4) can be described by the ODE (8). A detailed proof of Theorem 4.1 is shown in the supplemental material.

**Theorem 4.1.** *If Assumptions 1 and 2 hold, the sequence $\{\hat{\eta}_l\}_{l \geq 0}$ in Step 11 of Algorithm 1 converges to a connected internally chain recurrent set of* (8) *as $l \to \infty$ with probability 1.*

By definition of $\eta$ in (8), we know $\frac{\partial \eta(t)}{\partial t} = \frac{\partial \boldsymbol{v}}{\partial t} \cdot \text{Cov}_{\boldsymbol{v}}[\Gamma(\theta)]$. Since $\text{Cov}_{\boldsymbol{v}}[\Gamma(\theta)]$ is invertible by Assumption 1, (8) can be rewritten with variable $\boldsymbol{v}$

$$\frac{\partial \boldsymbol{v}}{\partial t} = \left( \tilde{L}(\boldsymbol{v}; \rho) \right)^{\mathsf{T}} \left( \text{Cov}_{\boldsymbol{v}}[\Gamma(\theta)] \right)^{-1}. \tag{9}$$

The conclusion of Theorem 4.1 can be equivalently stated in terms of the variable $\boldsymbol{v}$: the sequence $\{\boldsymbol{v}_l\}_{l \geq 0}$ of Algorithm 1 converges to a connected internally chain recurrent set of (9) as $l \to \infty$ with probability 1.

Intuitively, a point $\boldsymbol{v}_0 \in \mathcal{V}$ is *chain recurrent* for (9) if the solution $\boldsymbol{v}(t)$ of (9) with initial condition $\boldsymbol{v}(0) = \boldsymbol{v}_0$ can return to $\boldsymbol{v}_0$ within some finite time $t' > 0$ itself or just with finitely many arbitrarily small perturbations. An *internally chain recurrent set* is a nonempty compact *invariant* set of chain-recurrent points, i.e., $\boldsymbol{v}$ can never leave an internally chain recurrent set if $\boldsymbol{v}_0$ belongs to it.

Theorem 4.1 implies that with probability 1, the set of points that occur infinitely often in $\{\boldsymbol{v}_l\}_{l \geq 0}$ are internally chain recurrent for (9). Since $f_{\boldsymbol{v}}$ belongs to NEF, $\text{Cov}_{\boldsymbol{v}}[\Gamma(\theta)]$ is the Fisher information matrix at $\boldsymbol{v}$ and the right hand side of (9) is an estimate of the natural gradient of $\log L(\boldsymbol{v}; p)$ with a fixed indicator function $S$. This suggests that $\boldsymbol{v}$ evolves to increase $L(\boldsymbol{v}; \rho)$, which is consistent with the optimization problem (1) and our motivation to solve a constrained RL problem. Note that internally chain-recurrent sets are generally not unique and our algorithm can still converge to a local optimum.

To further interpret Theorem 4.1, we first note that any equilibrium of (8) forms an internally chain recurrent set by itself. The following result shows a sufficient condition for an equilibrium point $\bar{\boldsymbol{v}}^*$ of (8) to be locally asymptotically stable, i.e., there is a small neighborhood of this equilibrium of $\bar{\boldsymbol{v}}^*$ such that once entered, (9) will converge to $\bar{\boldsymbol{v}}^*$.

**Theorem 4.2.** *Let $\varphi : \mathcal{V} \to \mathbb{R}$ be any function such that $\frac{\partial}{\partial \boldsymbol{v}}\varphi(\boldsymbol{v}) = \tilde{L}(\boldsymbol{v}; \rho)$. Any equilibrium $\bar{\boldsymbol{v}}^* \in int(\mathcal{V})$ of* (9) *that is an isolated local maximum of $\varphi(\boldsymbol{v})$ is locally asympotically stable.*

The proof of Theorem 4.2 is done by constructing a local Lyapunov function and is given in the supplemental material. It also shows that $\varphi(\boldsymbol{v})$ always decreases in the interior of $\mathcal{V}$ unless it hits a stationary point of (9), which suggests a stronger property of our algorithm as stated in Theorem 4.3.

**Theorem 4.3.** *If all equilibria of* (9) *are isolated, the sequence $\{\boldsymbol{v}_l\}_{l \geq 0}$ derived by Algorithm 1 converges toward an equilibrium of* (9) *as $l \to \infty$ with probability 1.*

## 5 Experimental Results

We consider a mobile robot navigation task with only local sensors. There is a compact goal region $\mathcal{G}$ and a non-overlapping compact bad region $\mathcal{B}$ in the robot's environment. The transition function is deterministic. The robot uses a local sensing model to observe if $\mathcal{B}$ or $\mathcal{G}$ is in its neighborhood and the direction of the center of $\mathcal{G}$ in its local coordinate. Details of this experiment and the local sensing model can be found in the supplemental material.

We compare the performance of CCE to trust region policy optimization (TRPO) [27], a state-of-the-art unconstrained RL algorithm, and its variant for constrained problems, i.e., CPO [1]. For all

Objective value $G_{J_i}(\pi_\theta)$

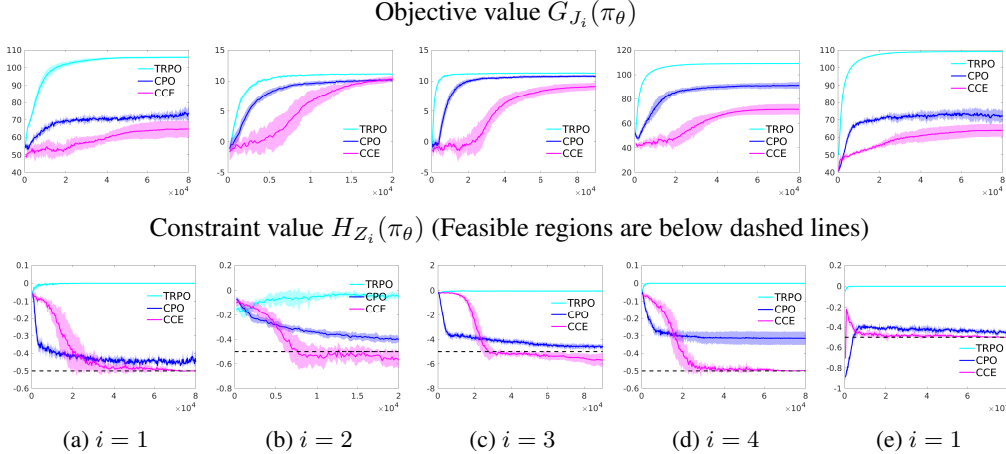

Constraint value $H_{Z_i}(\pi_\theta)$ (Feasible regions are below dashed lines)

(a) $i = 1$    (b) $i = 2$    (c) $i = 3$    (d) $i = 4$    (e) $i = 1$

Figure 1: Learning curves of CCE, CPO and TRPO with different objectives $G_{J_i}$ and constraints $H_{Z_i}$. The x-axes show the total number of sample trajectories for CCE and the total number of equivalent sample trajectories for TRPO and CPO. The y-axes show the sample mean of the objective and constraint values of the learned policy (for TRPO and CPO) or the learned policy distribution (for CCE). Each experiment is repeated for 5 times. More details can be found in the supplemental material.

Table 1: $J_i(\tau)$, $Z_i(\tau)$ and constraint upper bound $d_i$ for $i = 1, 2, 3, 4$, $\tau \in (\mathcal{S} \times \mathcal{A})^N$.

| $i$ | $J_i(\tau)$ | $Z_i(\tau)$ | $d_i$ | $J_i$ Markovian | $Z_i$ Markovian |
|---|---|---|---|---|---|
| 1 | 1 for each state in $\mathcal{G}$; $2\|y\|$ for each state with $y \in [-2, -0.2]$; 0 otherwise. | -1 if the robot arrives $\mathcal{G}$ which is absorbing; 0 otherwise. | -0.5 | Yes | Yes |
| 2 | 30 times the minimum signed distance from any state in $\tau$ to $\mathcal{B}$. | -1 if the robot visited $\mathcal{G}$ in $\tau$; 0 otherwise. | -0.5 | No | No |
| 3 | Same as $J_2(\tau)$. | -1 for each state in $\mathcal{G}$; 0 otherwise. | -5 | No | Yes |
| 4 | Same as $J_1(\tau)$. | -1 if the robot visits $\mathcal{G}$ and never visits $\mathcal{B}$; 0 otherwise. | -0.5 | Yes | No |

experiments, the agent's policy is modeled as a fully connected neural network with two hidden layers with 30 nodes in each layer. Trajectory length for all experiments is set to $N = 30$. All experiments are implemented in rllab [5].

Figure 1 shows the learning curves of CCE, TRPO and CPO for four different objectives and constraints ($i = 1, 2, 3, 4$). The objective functions and constraint functions used in each experiment are interpreted in Table 1. For experiments in which $J_i$ is not strictly positive, we use $\exp(J_i)$ instead of $J_i$ for the CCE. TRPO results show that the constraints cannot be satisfied by merely optimizing the corresponding objectives. We first initialize each experiment with a randomly generated infeasible policy. We find that CCE successfully outputs feasible policies in all

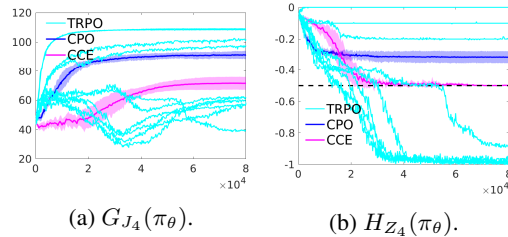

(a) $G_{J_4}(\pi_\theta)$.    (b) $H_{Z_4}(\pi_\theta)$.

Figure 2: Average performance of CCE, CPO and TRPO for Experiment 4 with initial feasible policy.

experiments. On the other hand, CPO needs significantly more samples to find a single feasible policy, or simply converges to an infeasible policy especially if the constraint is non-Markovian.

We repeat the first experiment ($i = 1$) with feasible initial policies and obtain the result in the last column of Figure 1. In this case, CPO leaves the feasible region rapidly and then follows generally the same path as if it is initialized with an infeasible policy. This behavior suggests that its incapability to enforce constraint satisfaction is not due to the lack of initial feasibility. Although CCE also leaves the feasible region at an early stage of iterations, it regains feasibility much faster than the previous case with infeasible initial polices. These results suggest that CCE is more reliable than CPO for applications where the strict constraint satisfaction is critical.

In Figure 2, we compare the performance of CPO and CCE in Experiment 4 to that of TRPO with objective $G_{J_4} - 100H_{Z_4}$. Due to the non-Markovian nature of $Z_4$, $H_{Z_4}(\pi)$ is not sensitive to local changes in $\pi(s)$ at any state $s$. It therefore makes it more difficult for standard RL algorithms to improve its $H_{Z_4}$-value. The fixed penalty coefficient 100 is chosen to be neither too large nor too small so it can show a large variety of locally optimal behaviors with very different $G_{J_4}$-values and $H_{Z_4}$-values. Figure 2 clearly shows the trade-off between $G_{J_4}$-values and $H_{Z_4}$-values, which partially explains the gap between $G_{J_4}$-value outputs of CCE and CPO. With a fixed penalty coefficient, the policies learned by TRPO are either infeasible or with very small constraint values. The policy output by CCE has higher $G_{J_4}$-value than all feasible policies found by TRPO and CPO.

## 6    Conclusions and Future Work

In this work, we studied a safe reinforcement learning problem with the constraints that are defined as the expected cost over finite-length trajectories. We proposed a constrained cross-entropy-based method to solve this problem, analyzed its asymptotic performance using an ODE and proved its convergence. We showed with simulation experiments that our method can effectively learn feasible policies without assumptions on the feasibility of initial policies with both Markovian and non-Markovian objective functions and constraint functions.

CCE is expected to be less sample-efficient than gradient-based methods especially for high-dimensional systems. Unlike gradient-based methods such as TRPO, CCE does not infer the performances of unseen policies from previous experience. As a result, it has to repetitively sample good policies in order to make steady improvement. Meanwhile, CCE can be easily parallelized as each sampled policy is evaluated independently. This may mitigate the problem of high sample complexity as other evolutionary methods [26].

Given all these limitations, we find the CCE method to be particularly useful in learning hierarchical policies. With a high-level policy that specifies intermediate goals and thus reduces the state space for low-level policies, we can use CCE to train a (locally) optimal low-level policy while satisfying local constraints. As shown in the experiment of our paper, CCE converges with reasonable sample complexity and outperforms CPO on its constraint performance. Since the satisfaction of low-level constraints is of critical significance to the performance of the overall policy, CCE seems to be especially well-suited for this application. In future work, we will combine this method with off-policy policy evaluation techniques such as [13; 9] to improve sample complexity.

## Acknowledgement

This work was supported in part by ONR N000141712623, ARO W911NF-15-1-0592 and DARPA W911NF-16-1-0001.

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
