[Supplementary Material]

# Supplemental material for the paper: Constrained Cross-Entropy Method for Safe Reinforcement Learning

We repeat some notations and definitions here and restate Problem 1.

For $\rho \in (0, 1)$, $\boldsymbol{v} \in \mathcal{V}$ and any function $X : \Theta \to \mathbb{R}$, we denote the $\rho$-quantile of $X$ for $\theta \sim f_{\boldsymbol{v}}$ by $\xi_X(\rho, \boldsymbol{v})$. We also define $\delta : \mathbb{R} \times \{\geq, \leq, >, <, =\} \times \mathbb{R} \to \{0, 1\}$ as an indicator function such that for $\circ \in \{\geq, \leq, >, <, =\}$, $\delta(x \circ y) = 1$ if and only if $x \circ y$ holds. The surrogate objective function for the unconstrained CE method is $\mathbb{E}_{\theta \sim f_{\boldsymbol{v}}(\cdot)}[G(\pi_\theta)\delta(G(\pi_\theta) \geq \xi_G(1 - \rho, \boldsymbol{v}))]$. In other words, a policy $\pi_\theta$ is considered as highly ranked if $G(\pi_\theta) \geq \xi_G(1 - \rho, \boldsymbol{v})$. When there is a constraint $H(\pi) \leq d$, we define $U : \Pi_\Theta \to \mathbb{R}$ such that $U(\pi_\theta) := G(\pi_\theta)\delta(H(\pi_\theta) \leq d)$ for any $\theta \in \Theta$ and extend the surrogate function as follows:

$$L(\boldsymbol{v}; \rho) := \begin{cases} \mathbb{E}_{\theta \sim f_{\boldsymbol{v}}(\cdot)}[G(\pi_\theta)\delta(H(\pi_\theta) \leq \xi_H(\rho, \boldsymbol{v}))], & \text{if } \xi_H(\rho, \boldsymbol{v}) > d; \\ \mathbb{E}_{\theta \sim f_{\boldsymbol{v}}(\cdot)}[U(\pi_\theta)\delta(U(\pi_\theta) \geq \xi_U(1 - \rho, \boldsymbol{v}))], & \text{otherwise.} \end{cases} \tag{1}$$

We can combine the two cases. Define $S : \Pi_\Theta \times \mathcal{V} \times (0, 1) \to \{0, 1\}$ such that

$$\begin{aligned} S(\pi_\theta, \boldsymbol{v}, \rho) := & \delta(\xi_H(\rho, \boldsymbol{v}) > d)\delta(H(\pi_\theta) \leq \xi_H(\rho, \boldsymbol{v})) + \\ & \delta(\xi_H(\rho, \boldsymbol{v}) \leq d)\delta(H(\pi_\theta) \leq d)\delta(U(\pi_\theta) \geq \xi_U(1 - \rho, \boldsymbol{v})), \end{aligned}$$

then (1) can be rewritten as

$$L(\boldsymbol{v}; \rho) = \mathbb{E}_{\theta \sim f_{\boldsymbol{v}}(\cdot)}[G(\pi_\theta)S(\pi_\theta, \boldsymbol{v}, \rho)]. \tag{2}$$

We first restate Problem 1 here.

**Problem 1.** *Given a set $\Pi = \{\pi_\theta : \theta \in \Theta\}$ of policies with parameter space $\Theta$, an NEF $F_\mathcal{V} = \{f_{\boldsymbol{v}}(\cdot) \in \mathcal{D}(\Theta) : \boldsymbol{v} \in \mathcal{V}\}$ of distributions over $\Theta$, two functions $G : \Pi \to \mathbb{R}^+$ and $H : \Pi \to \mathbb{R}$, a constraint upper bound $d$ and $\rho \in (0, 1)$, compute $\boldsymbol{v}^* \in \mathcal{V}$ such that*

$$\boldsymbol{v}^* = \arg\max_{\boldsymbol{v} \in \mathcal{V}} L(\boldsymbol{v}; \rho),$$

*where $L : \mathcal{V} \times (0, 1) \to \mathbb{R}$ is defined in* (2).

*Remark* 1. For technical reasons, we approximate the binary function $\delta$ with a Lipschitz continuous piecewise linear function $\tilde{\delta}_\varepsilon : \mathbb{R} \times \{\geq, \leq, >, <, =\} \times \mathbb{R} \to [0, 1]$ with $\varepsilon > 0$. For example, for any $(x, y) \in \mathbb{R}^2$:

$$\tilde{\delta}_\varepsilon(x \geq y) = \begin{cases} 1, & \text{if } x \geq y \\ \frac{x-y}{\varepsilon}, & \text{if } y > x \geq y - \varepsilon \\ 0, & \text{otherwise.} \end{cases}$$

With slight abuse of notation, we use $\delta$ to represent $\tilde{\delta}_\varepsilon$ for some small enough $\varepsilon$. Therefore $\delta(x \circ y)$ is Lipschitz continuous both in $x$ and in $y$.

## Proofs for Section 4.2

In this section we provide proofs for all theorems in our paper. The main idea behind the proof of Theorem 4.1 is similar to that of Theorem 3.1 Hu et al. [2012], although the details are adapted to our problem.

The following lemma shows that Assumption 1 in Section 4.2 is sufficient to guarantee that $m^{-1}$ exists and is continuously differentiable.

**Lemma 1.** *Let $F_{\mathcal{V}}$ be an NEF. Define $m : \mathbb{R}^{d_v} \to \mathbb{R}^{d_v}$ such that for each $\boldsymbol{v} \in \mathcal{V} := \{\boldsymbol{v} \in \mathbb{R}^{d_v} : K(\boldsymbol{v}) < \infty\}$, $m(\boldsymbol{v}) := \mathbb{E}_{\boldsymbol{v}}[\Gamma(\theta)]$. Then $m^{-1}$ exists and is continuously differentiable over $\{\eta : \exists \, \boldsymbol{v} \in int(\mathcal{V}) \; s.t. \; \eta = m(\boldsymbol{v})\}$.*

*Proof.*

$$\frac{\partial}{\partial \boldsymbol{v}} m(\boldsymbol{v}) = \frac{\partial}{\partial \boldsymbol{v}} \int_{\Theta} \Gamma(\theta) f_{\boldsymbol{v}}(\theta) d\theta$$

$$= \int_{\Theta} \left( \frac{\partial}{\partial \boldsymbol{v}} f_{\boldsymbol{v}}(\theta) \right) (\Gamma(\theta))^{\mathsf{T}} d\theta$$

$$= \int_{\Theta} \left( f_{\boldsymbol{v}}(\theta)(\Gamma(\theta) - m(\boldsymbol{v})) \right) (\Gamma(\theta))^{\mathsf{T}} d\theta$$

$$= \int_{\Theta} f_{\boldsymbol{v}}(\theta) \Gamma(\theta) \Gamma(\theta)^{\mathsf{T}} d\theta - m(\boldsymbol{v}) \cdot \left( \int_{\Theta} f_{\boldsymbol{v}}(\theta) \Gamma(\boldsymbol{v}) d\boldsymbol{v} \right)^{\mathsf{T}}$$

$$= \mathbb{E}_{\boldsymbol{v}}[(\Gamma(\theta) - m(\boldsymbol{v}))(\Gamma(\theta) - m(\boldsymbol{v}))^{\mathsf{T}}]$$

$$= \mathrm{Cov}_{\boldsymbol{v}}[\Gamma(\theta)],$$

which is positive definite by Assumption 1. As both $f_{\boldsymbol{v}}(\cdot)$ and $m(\boldsymbol{v})$ are continuous at $\boldsymbol{v} \in int(\mathcal{V})$, $m(\boldsymbol{v})$ is continuously differentiable with nonzero derivative at any $\boldsymbol{v} \in int(\mathcal{V})$. Therefore by inverse mapping theorem, $m^{-1}$ exists and is continuously differentiable over $\{\eta : \exists \boldsymbol{v} \in int(\mathcal{V}) s.t. \eta = m(\boldsymbol{v})\}$. $\qquad\square$

The following lemma derives (5) in Section 4.2.

**Lemma 2.** *Let $F_{\mathcal{V}}$ be an NEF, $\boldsymbol{v} \in \mathcal{V}$ and $g \in \mathcal{D}(\Theta)$. Let $\boldsymbol{v}' := \arg\min_{\boldsymbol{v} \in \mathcal{V}} D_{KL}(\tilde{g}_{\boldsymbol{v}}, f_{\boldsymbol{v}})$. Assume that $\boldsymbol{v}' \in int(\mathcal{V})$, then*

$$m(\boldsymbol{v}') - m(\boldsymbol{v}) = -\alpha \left( \frac{\partial}{\partial \boldsymbol{v}''} D_{KL}(g_{\boldsymbol{v}}^*, f_{\boldsymbol{v}''}) \right) \Big|_{\boldsymbol{v}''=\boldsymbol{v}}.$$

*Proof.* As $F_{\mathcal{V}}$ is an NEF, $\log f_{\boldsymbol{v}''}(\theta) = (\boldsymbol{v}'')^{\mathsf{T}} \Gamma(\theta) - K(\boldsymbol{v}'')$ is concave in $\boldsymbol{v}''$ and thus $-\int_{\Theta} \tilde{g}_{\boldsymbol{v}}(\theta) \log f_{\boldsymbol{v}''}(\theta) d\theta$ is convex in $\boldsymbol{v}''$. Therefore $\boldsymbol{v}'$ can be found by setting the gradient to zero.

Let $-\frac{\partial}{\partial \boldsymbol{v}''} \int_{\Theta} \tilde{g}_{\boldsymbol{v}}(\theta) \log f_{\boldsymbol{v}''}(\theta) d\theta = \boldsymbol{0}$. By Assumption (2d), $\Theta$ is bounded. By definition of NEF, $f_{\boldsymbol{v}''}$ is continuously differentiable. Then by dominated convergent theorem,

$$- \frac{\partial}{\partial \boldsymbol{v}''} \int_{\Theta} \tilde{g}_{\boldsymbol{v}}(\theta) \log f_{\boldsymbol{v}''}(\theta) d\theta$$

$$= - \int_{\Theta} \tilde{g}_{\boldsymbol{v}}(\theta) \frac{\partial}{\partial \boldsymbol{v}''} \log f_{\boldsymbol{v}''}(\theta) d\theta$$

$$= - \int_{\Theta} \tilde{g}_{\boldsymbol{v}}(\theta)(\Gamma(\theta) - m(\boldsymbol{v}'')) d\theta$$

$$= - \int_{\Theta} (\alpha g_{\boldsymbol{v}}^*(\theta) + (1 - \alpha) f_{\boldsymbol{v}}(\theta)) \Gamma(\theta) d\theta + m(\boldsymbol{v}'')$$

$$= - \alpha \left( \int_{\Theta} g_{\boldsymbol{v}}^*(\theta) \Gamma(\theta) d\theta - m(\boldsymbol{v}) \right) + m(\boldsymbol{v}'') - m(\boldsymbol{v})$$

$$= - \alpha \left( \mathbb{E}_{g_{\boldsymbol{v}}^*}[\Gamma(\theta)] - m(\boldsymbol{v}) \right) + m(\boldsymbol{v}'') - m(\boldsymbol{v}),$$

which equals to $\boldsymbol{0}$ when $\boldsymbol{v}'' = \boldsymbol{v}'$.

Since $G(\pi_\theta) \geq 0$ for all $\theta \in \Theta$, $G(\pi_\theta) \equiv |G(\pi_\theta)|$. Therefore

$$m(\boldsymbol{v}') = m(\boldsymbol{v}) + \alpha \left( \int_{\Theta} g_{\boldsymbol{v}}^*(\theta) \Gamma(\theta) d\theta - m(\boldsymbol{v}) \right)$$

$$= (1 - \alpha) m(\boldsymbol{v}) + \alpha \mathbb{E}_{g_{\boldsymbol{v}}^*}[\Gamma(\theta)].$$

On the other hand,

$$\frac{\partial}{\partial \boldsymbol{v}'} D_{KL}(g_{\boldsymbol{v}}^*, f_{\boldsymbol{v}'})$$

$$= \frac{\partial}{\partial \boldsymbol{v}'} \mathbb{E}_{g_{\boldsymbol{v}}^*} \log \frac{g_{\boldsymbol{v}}^*(\theta)}{f_{\boldsymbol{v}'}(\theta)} \mid_{\boldsymbol{v}'=\boldsymbol{v}} = -\frac{\partial}{\partial \boldsymbol{v}'} \mathbb{E}_{g_{\boldsymbol{v}}^*} \log f_{\boldsymbol{v}'}(\theta) \mid_{\boldsymbol{v}'=\boldsymbol{v}}$$

$$= -\frac{\partial}{\partial \boldsymbol{v}'} \mathbb{E}_{g_{\boldsymbol{v}}^*} \big( \boldsymbol{v}'^{\mathsf{T}} \Gamma(\theta) - K(\boldsymbol{v}') \big) \mid_{\boldsymbol{v}'=\boldsymbol{v}}$$

$$= -\frac{\partial}{\partial \boldsymbol{v}'} \big( \boldsymbol{v}'^{\mathsf{T}} \mathbb{E}_{g_{\boldsymbol{v}}^*}[\Gamma(\theta)] \big) \mid_{\boldsymbol{v}'=\boldsymbol{v}} + \frac{\partial}{\partial \boldsymbol{v}'} K(\boldsymbol{v}') \mid_{\boldsymbol{v}'=\boldsymbol{v}}$$

$$= -\mathbb{E}_{g_{\boldsymbol{v}}^*}[\Gamma(\theta)] + m(\boldsymbol{v}).$$

Therefore

$$m(\boldsymbol{v}') - m(\boldsymbol{v}) = -\alpha \Big( \frac{\partial}{\partial \boldsymbol{v}''} D_{KL}(g_{\boldsymbol{v}}^*, f_{\boldsymbol{v}''}) \Big) \Big|_{\boldsymbol{v}''=\boldsymbol{v}}.$$

$\square$

**Proof of Theorem 4.1**

In practice we can only estimate expectations and quantiles using finite samples. Let $\mathcal{Y}_l = \{\theta_1, \ldots, \theta_{n_l}\}$ be the set of samples in the $l^{th}$ iteration with sampling distribution $f_{\boldsymbol{v}_l}$. We denote the sample estimate of $S(\pi_\theta, \boldsymbol{v}, \rho)$ by $\hat{S}(\pi_\theta, \boldsymbol{v}, \rho)$.

Consider the equation in the Step 11 of Algorithm 1:

$$\hat{\eta}_{l+1} = \alpha_l \frac{\sum_{i=1}^{n_l} G(\pi_{\theta_i}) \hat{S}(\pi_{\theta_i}, \boldsymbol{v}_l, \rho) \Gamma(\theta_i)}{\sum_{i=1}^{n_l} G(\pi_{\theta_i}) \hat{S}(\pi_{\theta_i}, \boldsymbol{v}_l, \rho)} + (1 - \alpha_l) \Big( \frac{\lambda_l}{n_l} \sum_{i=1}^{n_l} \Gamma(\theta_i) + (1 - \lambda_l) \hat{\eta}_l \Big), \quad (3)$$

where $\boldsymbol{v}_l = m^{-1}(\hat{\eta}_l)$.

We need to show the connection between (3) and the ODE (8) in Section 4.3. The first step is to rewrite (3) to explicitly compare the sampling-based estimates to their true values. Or equivalently,

$$\hat{\eta}_{l+1} - \hat{\eta}_l$$

$$= \alpha_l \Big( \frac{\sum_{i=1}^{n_l} G(\pi_{\theta_i}) \hat{S}(\pi_{\theta_i}, \boldsymbol{v}_l, \rho) \Gamma(\theta_i)}{\sum_{i=1}^{n_l} G(\pi_{\theta_i}) \hat{S}(\pi_{\theta_i}, \boldsymbol{v}_l, \rho)} - \hat{\eta}_l \Big) + (1 - \alpha_l) \lambda_l \Big( \frac{1}{n_l} \sum_{i=1}^{n_l} \Gamma(\theta_i) - \hat{\eta}_l \Big)$$

$$= \alpha_l \Big( \frac{\mathbb{E}_{\boldsymbol{v}_l}[G(\pi_\theta) S(\pi_\theta, \boldsymbol{v}_l, \rho) \Gamma(\theta)]}{\mathbb{E}_{\boldsymbol{v}_l}[G(\pi_\theta) S(\pi_\theta, \boldsymbol{v}_l, \rho)]} - \hat{\eta}_l \Big)$$

$$+ \alpha_l \Big( \frac{\sum_{i=1}^{n_l} G(\pi_{\theta_i}) \hat{S}(\pi_{\theta_i}, \boldsymbol{v}_l, \rho) \Gamma(\theta_i)}{\sum_{i=1}^{n_l} G(\pi_{\theta_i}) \hat{S}(\pi_{\theta_i}, \boldsymbol{v}_l, \rho)} - \frac{\mathbb{E}_{\boldsymbol{v}_l}[G(\pi_\theta) S(\pi_\theta, \boldsymbol{v}_l, \rho) \Gamma(\theta)]}{\mathbb{E}_{\boldsymbol{v}_l}[G(\pi_\theta) S(\pi_\theta, \boldsymbol{v}_l, \rho)]} \Big)$$

$$+ (1 - \alpha_l) \Big( \frac{\lambda_l}{n_l} \sum_{i=1}^{n_l} \Gamma(\theta_i) - \lambda_l \hat{\eta}_l \Big).$$

Define

$$L_l = \frac{\mathbb{E}_{\boldsymbol{v}_l}[G(\pi_\theta) S(\pi_\theta, \boldsymbol{v}_l, \rho) \Gamma(\theta)]}{\mathbb{E}_{\boldsymbol{v}_l}[G(\pi_\theta) S(\pi_\theta, \boldsymbol{v}_l, \rho)]} - \hat{\eta}_l$$

$$b_l = \frac{\sum_{i=1}^{n_l} G(\pi_{\theta_i}) \hat{S}(\pi_{\theta_i}, \boldsymbol{v}_l, \rho) \Gamma(\theta_i)}{\sum_{i=1}^{n_l} G(\pi_{\theta_i}) \hat{S}(\pi_{\theta_i}, \boldsymbol{v}_l, \rho)} - \frac{\mathbb{E}_{\boldsymbol{v}_l}[G(\pi_\theta) S(\pi_\theta, \boldsymbol{v}_l, \rho) \Gamma(\theta)]}{\mathbb{E}_{\boldsymbol{v}_l}[G(\pi_\theta) S(\pi_\theta, \boldsymbol{v}_l, \rho)]} \qquad (4)$$

$$w_l = \frac{1 - \alpha_l}{\alpha_l} \Big( \frac{\lambda_l}{n_l} \sum_{i=1}^{n_l} \Gamma(\theta_i) - \lambda_l \hat{\eta}_l \Big),$$

then (3) can be rewritten as

$$\hat{\eta}_{l+1} - \hat{\eta}_l = \alpha_l \Big( L_l + b_l + w_l \Big). \qquad (5)$$

Note that the first term in $L_l$ coincides with $\mathbb{E}_{\boldsymbol{v}_l^*}[\Gamma(\theta)]$ where $g_{\boldsymbol{v}_l}^*$ is defined in Equation (3) of Section 4.2. It holds by (6) in Section 4.2 that

$$
\begin{aligned}
L_l &= \mathbb{E}_{\boldsymbol{v}_l^*}[\Gamma(\theta)] - \hat{\eta}_l \\
&= \mathbb{E}_{\boldsymbol{v}_l^*}[\Gamma(\theta)] - m(\boldsymbol{v}_l) \\
&= \frac{\partial}{\partial \boldsymbol{v}'} \log \mathbb{E}_{\boldsymbol{v}'}[G(\pi_\theta) S(\pi_\theta, \boldsymbol{v}_l, \rho)] \Big|_{\boldsymbol{v}' = \boldsymbol{v}_l},
\end{aligned}
$$

where the right hand side is the same as that of the ODE (7) in Section 4.2.

We aim to show the connection between $\{\eta_l\}_{l \geq 0}$ (as defined in (3) or (5)) and the ODE (8) using the following conclusion in stochastic approximation.

**Theorem 1.** *(Theorem 1.2, Benaim [1996]) Let $H : \mathbb{R}^m \to \mathbb{R}^m$ be a continuous vectorfield with unique integral curves. Let $\{w_n\}_{n \geq 0}$ be the solution to $w_{n+1} - w_n = \gamma_n(H(w_n) + u_n + b_n)$, where $\{\gamma_n\}_{n \geq 0}$ is a decreasing gain sequence. Assume that*

- *$\{\gamma_n\}_{n \geq 0}$ is bounded.*

- *$\lim_{n \to +\infty} b_n = 0$.*

- *For each $T > 0$,*

$$
\lim_{n \to \infty} \left( \sup_{k:0 \leq \tau_k - \tau_n \leq T} \left\| \sum_{i=n}^{k-1} \gamma_i u_i \right\| \right) = 0.
$$

*Then the limit set of $\{w_n\}_{n \geq 0}$ is a connected set internally chain-recurrent for the flow induced by $H$.*

We first show that $\lim_{l \to \infty} b_l = 0$ where $b_l$ is defined in (4).

**Lemma 3.** *Given Assumption (2b), (2c), (2d), (2e), $\lim_{l \to \infty} b_l = 0$, w.p.1.*

In order to prove Lemma 3, we first show that the sample quantile is an unbiased estimate of the true quantile, which is stated in Lemma 4. Although we show the result for $H$, similar results apply for $U$.

**Lemma 4.** *Let $\xi(\rho, \boldsymbol{v}_l)$ be the true $(1 - \rho)$-quantile of $H(\pi_\theta)$ with $\theta \sim f_{\boldsymbol{v}_l}$ and $\hat{\xi}_l$ be a sample $(1 - \rho)$-quantile acquired from $n_l$ i.i.d. samples. Given Assumption (2b), (2c), (2e), $\hat{\xi}_l - \xi(\rho, \boldsymbol{v}_l) \to 0$ as $l \to \infty$ w.p.1.*

*Proof.* By Assumption (2e), $H(\pi) \in \mathcal{H} := [H_{min}, H_{max}]$ for all $\pi \in \Pi$. It can be verified that any true $(1 - \rho)$-quantile $\xi(\rho, \boldsymbol{v}_l)$ with $\theta \sim f_{\boldsymbol{v}_l}(\cdot)$ is an optimal solution of the following optimization problem Homem-de Mello [2007]:

$$
\min_{\gamma \in \mathcal{H}} J_l(\gamma) := \mathbb{E}_{\boldsymbol{v}_l}[h(H(\pi_\theta), \gamma)]
$$

$$
s.t. \ h(H(\pi_\theta), \gamma) = \begin{cases} (1 - \rho)(H(\pi_\theta) - \gamma), & \text{if } H(\pi_\theta) \geq \gamma, \\ \rho(\gamma - H(\pi_\theta)), & \text{if } H(\pi_\theta) < \gamma. \end{cases}
$$

Similarly the sample $(1 - \rho)$-quantile $\hat{\xi}_l$ can be computed by minimizing

$$
\hat{J}_l(\gamma) := \frac{1}{n_l} \sum_{i=1}^{n_l} h(H(\pi_{\theta_i}), \gamma),
$$

where $\{\theta_1, \ldots, \theta_{n_l}\}$ are i.i.d. samples of distribution $f_{\boldsymbol{v}_l}$.

We first show that $J_l(\gamma)$ uniformly converges to $\hat{J}_l(\gamma)$ over $\mathcal{H}$ w.p.1, i.e. $\sup_{\gamma \in \mathcal{H}} |J_l(\gamma) - \hat{J}_l(\gamma)| \to 0$ as $l \to \infty$ w.p.1.

Let $\delta$ and $r$ be two arbitrary scalars such that $\delta > 0$ and $r \leq \frac{\delta}{3 \max(\rho, 1 - \rho)}$. Let $B(\gamma, r) := \{\gamma' \in \mathcal{H} : \|\gamma - \gamma'\| \leq r\}$ be the $r$-neighborhood of $\gamma \in \mathcal{H}$ within $\mathcal{H}$. Since $\mathcal{H}$ is compact, there exists a finite

cover $\mathcal{U} = \{h_1, \ldots, h_k\} \subset \mathcal{H}$ of $\mathcal{H}$ such that $\mathcal{H} \subseteq \bigcup_{i=1}^{k} B(h_i, r)$. For each $\gamma \in \mathcal{H}$, let $h(\gamma) \in \mathcal{U}$ be the closest component in $\mathcal{H}$. By definition, $\sup_{\gamma \in \mathcal{H}} ||\gamma - h(\gamma)|| \leq r$. For any $\gamma \in \mathcal{H}$,

$$|J_l(\gamma) - J_l(h(\gamma))|$$
$$= |\mathbb{E}_{\boldsymbol{v}_l}[h(H(\pi_\theta), \gamma)] - \mathbb{E}_{\boldsymbol{v}_l}[h(H(\pi_\theta), h(\gamma))]|$$
$$\leq \max(\rho, 1 - \rho)||\gamma - h(\gamma)|| \leq \frac{\delta}{3}.$$

$$|\hat{J}_l(\gamma) - \hat{J}_l(h(\gamma))|$$
$$= \frac{1}{n_l} |\sum_{i=1}^{n_l} \Big( h(H(\pi_{\theta_i}), \gamma) - h(H(\pi_{\theta_i}), h(\gamma)) \Big)|$$
$$\leq \max(\rho, 1 - \rho)||\gamma - h(\gamma)|| \leq \frac{\delta}{3}.$$

As $H(\cdot) \subseteq [H_{min}, H_{max}]$, we can bound the probability that $|J_l(h(\gamma)) - \hat{J}_l(h(\gamma))| > \delta$ for any $\delta \geq 0$ by Hoeffding's inequality:

$$Pr\big(|J_l(h(\gamma)) - \hat{J}_l(h(\gamma))| \geq \frac{\delta}{3}\big)$$
$$\leq 2 \exp(-\frac{2n_l \delta^2}{9|H_{max} - H_{min}|^2}).$$

As $\mathrm{card}(\mathcal{U}) = k < \infty$, we can bound the probability that $|J_l(h_i) - \hat{J}_l(h_i)| < \frac{\delta}{3}$ holds for all $h_i \in \mathcal{U}$ with the union bound:

$$Pr\big((\max_{h_i \in \mathcal{U}} |J_l(h_i) - \hat{J}_l(h_i)|) \geq \frac{\delta}{3}\big)$$
$$\leq \sum_{i=1}^{k} Pr\big(|J_l(h_i) - \hat{J}_l(h_i)| \geq \frac{\delta}{3}\big)$$
$$\leq 2k \exp(-\frac{2n_l \delta^2}{9|H_{max} - H_{min}|^2}).$$

Therefore with probability at least $(1 - 2k \exp(-\frac{2n_l \delta^2}{9|H_{max} - H_{min}|^2}))$,

$$|J_l(\gamma) - \hat{J}_l(\gamma)| \leq \frac{\delta}{3} + \frac{\delta}{3} + \frac{\delta}{3} = \delta$$

holds uniformly for all $\gamma \in \mathcal{H}$. In other words,

$$Pr(\sup_{\gamma \in \mathcal{H}} |J_l(\gamma) - \hat{J}_l(\gamma)| > \delta)$$
$$\leq 2k \exp(-\frac{2n_l \delta^2}{9|H_{max} - H_{min}|^2})).$$

Therefore

$$\sum_{l=1}^{\infty} Pr(\sup_{\gamma \in \mathcal{H}} |J_l(\gamma) - \hat{J}_l(\gamma)| > \delta)$$
$$\leq \sum_{l=1}^{\infty} 2k \exp(-\frac{2n_l \delta^2}{9|H_{max} - H_{min}|^2})) < \infty.$$

By Assumption (2b), the last inequality holds as $n_l = \Theta(l^\beta)$ with $\beta > 0$. By Borel-Cantelli Lemma, $Pr(\sup_{\gamma \in \mathcal{H}} |J_l(\gamma) - \hat{J}_l(\gamma)| > \delta \ i.o.) = 0$. As the above proof holds for any $\delta > 0$, $\sup_{\gamma \in \mathcal{H}} |J_l(\gamma) - \hat{J}_l(\gamma)| \to 0$ as $l \to \infty$ w.p.1. In other words, $\hat{J}_l(\cdot)$ converges uniformly to $J_l(\cdot)$ as $l \to \infty$ w.p.1. Note that this uniform convergence holds whenever Assumption (2b) and (2e) hold.

Then we prove that $\lim_{l \to +\infty} |\hat{\xi}_l - \xi(\rho, \boldsymbol{v}_l)| = 0$, w.p.1.

Since $\sup_{\gamma \in \mathcal{H}} |J_l(\gamma) - \hat{J}_l(\gamma)| \to 0$ as $l \to \infty$ w.p.1, there exists some $L(\varepsilon) > 0$ for any $\varepsilon > 0$ such that $\sup_{\gamma \in \mathcal{H}} |J_l(\gamma) - \hat{J}_l(\gamma)| < \varepsilon$ holds for any $l > L(\varepsilon)$, w.p.1. Therefore with probability 1 and $l > L(\varepsilon)$,

$$J_l(\hat{\xi}_l) - \varepsilon < \hat{J}_l(\hat{\xi}_l), \quad \hat{J}_l(\xi(\rho, \boldsymbol{v}_l)) < J_l(\xi(\rho, \boldsymbol{v}_l)) + \varepsilon.$$

By definition of $\xi(\rho, \boldsymbol{v}_l)$ and $\hat{\xi}_l$, i.e., $\xi(\rho, \boldsymbol{v}_l)$ minimizes $J_l(\cdot)$ and $\hat{\xi}_l$ minimizes $\hat{J}_l(\cdot)$, we get

$$J_l(\xi(\rho, \boldsymbol{v}_l)) \leq J_l(\hat{\xi}_l), \quad \hat{J}_l(\hat{\xi}_l) \leq \hat{J}_l(\xi(\rho, \boldsymbol{v}_l)).$$

Combining the above two equalities, we get

$$J_l(\xi(\rho, \boldsymbol{v}_l)) - \varepsilon \leq J_l(\hat{\xi}_l) - \varepsilon < \hat{J}_l(\hat{\xi}_l)$$
$$\leq \hat{J}_l(\xi(\rho, \boldsymbol{v}_l)) < J_l(\xi(\rho, \boldsymbol{v}_l)) + \varepsilon.$$

Therefore with probability 1,

$$J_l(\xi(\rho, \boldsymbol{v}_l)) - \varepsilon < J_l(\hat{\xi}_l) < J_l(\xi(\rho, \boldsymbol{v}_l)) + \varepsilon$$

for sufficiently large $l$. In other words, $J_l(\hat{\xi}_l) - J_l(\xi(\rho, \boldsymbol{v}_l)) \to 0$ as $l \to +\infty$ w.p.1.

By Assumption (2c), the $(1 - \rho)$-quantile of $\{H(\pi_\theta) : \theta \sim f_{\boldsymbol{v}}(\cdot)\}$ is unique for all $\boldsymbol{v} \in \mathcal{V}$. Therefore $J_l(\gamma)$ is minimized with a unique $\xi(\rho, \boldsymbol{v}_l)$. As $J_l(\gamma)$ is also continuous in $\gamma$, for $\varepsilon_l > 0$ that is small enough, there exists $\delta_l(\varepsilon_l) > 0$ such that $|J_l(\gamma) - J_l(\xi(\rho, \boldsymbol{v}_l))| < \varepsilon_l$ if and only if $\|\xi(\rho, \boldsymbol{v}_l) - \gamma\| < \delta_l(\varepsilon_l)$. Moreover, $\delta_l(\varepsilon_l) \to 0^+$ as $\varepsilon_l \to 0^+$ for each $l$.

Assume that $\hat{\xi}_l - \xi(\rho, \boldsymbol{v}_l)$ does not converge to 0 w.p.1. Then $\exists \bar{\delta} > 0$ such that $Pr(\{|\hat{\xi}_l - \xi(\rho, \boldsymbol{v}_l)| > \bar{\delta} \text{ i.o.}\}) > 0$. With positive probability, there exists a subsequence $\{l_k\}_{k \geq 0} \in \mathbb{N}^\infty$ such that $|\hat{\eta}_{l_k} - \xi(\rho, \boldsymbol{v}_{l_k})| > \bar{\delta}$ for each $k \in \mathbb{N}$ and $\lim_{k \to \infty} J_{l_k}(\hat{\eta}_{l_k}) - J_{l_k}(\xi(\rho, \boldsymbol{v}_{l_k})) = 0$. We can further select a subsequence $\{l_{k_j}\}_{j \geq 0} \subset \{l_k\}_{k \geq 0}$ such that $|J_{l_{k_j}}(\hat{\eta}_{l_{k_j}}) - J_{l_{k_j}}(\xi(\rho, \boldsymbol{v}_{l_{k_j}}))| < \frac{1}{2^j}$ and $|\hat{\eta}_{l_{k_j}} - \xi(\rho, \boldsymbol{v}_{l_{k_j}})| > \bar{\delta}$. Since each $\xi(\rho, \boldsymbol{v}_{l_{k_j}})$ is unique, there exists $j' \in \mathbb{N}$ such that $\delta_{l_{k_{j'}}}(\frac{1}{2^{j'}}) < \bar{\delta}$, which contradicts our assumption that such a sequence $\{l_k\}_{k \geq 0}$ exists.

Therefore $\lim_{l \to +\infty} |\hat{\xi}_l - \xi(\rho, \boldsymbol{v}_l)| \to 0$ w.p.1. $\qquad\square$

We can now give a proof to Lemma 3.

*Proof.* By Assumption (2e), $\inf_{\pi \in \Pi} G(\pi) > 0$. By definition of $(1 - \rho)$-quantile, it holds for any $\boldsymbol{v} \in \mathcal{V}$ that

$$\mathbb{E}_{\boldsymbol{v}}[G(\pi_\theta)S(\pi_\theta, \boldsymbol{v}, \rho)] \geq \inf_{\pi \in \Pi} G(\pi)\rho > 0.$$

Similarly we can show

$$\sum_{i=1}^{n_l} G(\pi_{\theta_i})S(\pi_{\theta_i}, \boldsymbol{v}, \rho) \geq \inf_{\pi \in \Pi} G(\pi) > 0.$$

There are two types of approximation involved in $b_l$: the first is to approximate $\xi_H(\rho, \boldsymbol{v}_l)$ and $\xi_U(\rho, \boldsymbol{v}_l)$ by $\hat{\xi}_{H,l}$ and $\hat{\xi}_{U,l}$. The second is to approximate the expectation (e.g. $\mathbb{E}_{\boldsymbol{v}_l}[G(\pi_\theta)\hat{S}(\pi_{\theta_i}, \boldsymbol{v}_l, \rho)\Gamma(\theta)]$) with sample mean (e.g. $\frac{1}{n_l} \sum_{i=1}^{n_l} G(\pi_{\theta_i})\hat{S}(\pi_{\theta_i}, \boldsymbol{v}_l, \rho)\Gamma(\theta_i)$).

As we have shown that $\lim_{l \to \infty} |\xi_H(\rho, \boldsymbol{v}_l) - \hat{\xi}_{H,l}| = 0$ w.p.1 and $\lim_{l \to \infty} |\xi_U(\rho, \boldsymbol{v}_l) - \hat{\xi}_{U,l}| = 0$ w.p.1 by Lemma 4, we can also get $\lim_{l \to \infty} |S(\pi_\theta, \boldsymbol{v}_l, \rho) - \hat{S}(\pi_\theta, \boldsymbol{v}_l, \rho)| = 0$ w.p.1. We only need to consider the second part in this proof.

$\Gamma(\cdot)$ is bounded as it is a continuous function defined over a compact set (by Assumption (2d)). By Assumption (2e), both $G$ and $H$ are bounded over $\Pi$. By Remark 1, $\delta(x \circ y)$ is bounded (by 1, to be specific) and Lipschitz continuous in both $x$ and $y$. Let $M > 0$ be a constant such that $\sup_{\theta \in \Theta} |G(\pi_\theta)\Gamma(\theta)| \leq M$. Therefore $\lim_{l \to \infty} \left| \frac{1}{n_l} \sum_{i=1}^{n_l} G(\pi_{\theta_i})\hat{S}(\pi_{\theta_i}, \boldsymbol{v}_l, \rho)\Gamma(\theta_i) - \frac{1}{n_l} \sum_{i=1}^{n_l} G(\pi_{\theta_i})S(\pi_{\theta_i}, \boldsymbol{v}_l, \rho)\Gamma(\theta_i) \right| = 0$ w.p.1.

As $G(\pi_\theta)$, $S(\pi_\theta, \boldsymbol{v}_l, \rho)$, $\Gamma(\theta)$ are all bounded for any $\theta$ and $\rho$, there exist finite $a, b$ such that $a \leq G(\pi_\theta)S(\pi_\theta, \boldsymbol{v}_l, \rho)\Gamma(\theta) \leq b$ for any $\theta \in \Theta$. By Hoeffding's inequality, for any $\varepsilon > 0$

$$Pr\Big(\Big|\frac{1}{n_l}\sum_{i=1}^{n_l}G(\pi_{\theta_i})S(\pi_{\theta_i}, \boldsymbol{v}_l, \rho)\Gamma(\theta_i) - \mathbb{E}_{\boldsymbol{v}_l}[G(\pi_\theta)S(\pi_\theta, \boldsymbol{v}_l, \rho)\Gamma(\theta)]\Big| \geq \varepsilon\Big)$$

$$\leq 2\exp\Big(\frac{-2n_l\varepsilon^2}{(b-a)^2}\Big).$$

By Assumption (2b), $n_l = \Theta(l^\beta)$ and $\beta > 0$. Therefore for any $\varepsilon > 0$,

$$\sum_{l=1}^{\infty} Pr\Big(\Big|\frac{1}{n_l}\sum_{i=1}^{n_l}G(\pi_{\theta_i})S(\pi_{\theta_i}, \boldsymbol{v}_l, \rho)\Gamma(\theta_i) - \mathbb{E}_{\boldsymbol{v}_l}[G(\pi_\theta)S(\pi_\theta, \boldsymbol{v}_l, \rho)\Gamma(\theta)]\Big| \geq \varepsilon\Big)$$

$$\leq \sum_{l=1}^{\infty} 2\exp\Big(\frac{-2n_l\varepsilon^2}{(b-a)^2}\Big) < \infty.$$

Then by Borel-Cantelli Lemma,

$$\Big|\frac{1}{n_l}\sum_{i=1}^{n_l}G(\pi_{\theta_i})S(\pi_{\theta_i}, \boldsymbol{v}_l, \rho)\Gamma(\theta_i)$$

$$- \mathbb{E}_{\boldsymbol{v}_l}[G(\pi_\theta)S(\pi_\theta, \boldsymbol{v}_l, \rho)\Gamma(\theta)]\Big| \to 0, w.p.1.$$

Therefore

$$\frac{1}{n_l}\sum_{i=1}^{n_l}G(\pi_{\theta_i})S(\pi_{\theta_i}, \boldsymbol{v}_l, \rho)\Gamma(\theta_i) - \mathbb{E}_{\boldsymbol{v}_l}[G(\pi_\theta)S(\pi_\theta, \boldsymbol{v}_l, \rho)\Gamma(\theta)] \to 0$$

as $l \to \infty$ w.p.1.

We can show that $\frac{1}{n_l}\sum_{i=1}^{n_l}G(\pi_{\theta_i})S(\pi_{\theta_i}, \boldsymbol{v}_l, \rho) - \mathbb{E}_{\boldsymbol{v}_l}[G(\pi_\theta)S(\pi_\theta, \boldsymbol{v}_l, \rho)] \to 0$ as $l \to \infty$ w.p.1 in exactly the same way as above; it is a special case when $\Gamma(\theta) = 1$ for all $\theta \in \Theta$.

By continuous mapping theorem, the facts that with probability 1,

$$\mathbb{E}_{\boldsymbol{v}_l}[G(\pi_\theta)S(\pi_\theta, \boldsymbol{v}_l, \rho)] > 0, \ \forall \boldsymbol{v} \in \mathcal{V},$$

$$\sum_{i=1}^{n_l}G(\pi_{\theta_i})\hat{S}(\pi_{\theta_i}, \boldsymbol{v}_l, \rho) > 0,$$

$$\lim_{l\to\infty}\Big|\frac{1}{n_l}\sum_{i=1}^{n_l}G(\pi_{\theta_i})S(\pi_{\theta_i}, \boldsymbol{v}_l, \rho)\Gamma(\theta_i) - \mathbb{E}_{\boldsymbol{v}_l}[G(\pi_\theta)S(\pi_\theta, \boldsymbol{v}_l, \rho)\Gamma(\theta)]\Big| = 0,$$

$$\lim_{l\to\infty}\frac{1}{n_l}\sum_{i=1}^{n_l}G(\pi_{\theta_i})\hat{S}(\pi_{\theta_i}, \boldsymbol{v}_l, \rho) - \mathbb{E}_{\boldsymbol{v}_l}[G(\pi_\theta)S(\pi_\theta, \boldsymbol{v}_l, \rho)] = 0,$$

guarantee that $\lim_{l\to\infty} b_l = 0$, w.p.1.

$\square$

Now we restate Theorem 4.1 and provide a proof.

**Theorem 4.1.** *If Assumptions 1 - (2e) hold, the sequence $\{\hat{\eta}_l\}_{l\geq0}$ in Step 11 of Algorithm 1 converges to a connected internally chain recurrent set of (8) as $l \to \infty$ with probability 1.*

*Proof.* We connect the sequence $\{\hat{\eta}_l\}_{l\geq0}$ to the ODE (8) by applying Theorem 1. We need to verify that all sufficient conditions in 1 hold properly. By (5), $\hat{\eta}_{l+1} - \hat{\eta}_l = \alpha_l\big(L_l + b_l + w_l\big)$.

- By Assumption (2a), $\tilde{L}(\boldsymbol{v}; \rho)$ is continuous in $\boldsymbol{v} \in int(\mathcal{V})$. By Lemma 1, $m^{-1}(\eta)$ is continuous in $\eta$. Therefore $\tilde{L}(\boldsymbol{v}; \rho)\big|_{\boldsymbol{v}=m^{-1}(\eta)}$ is continuous in $\eta$. (8) has a unique integral curve by Assumption (2a).

- By Assumption (2b), $\{\alpha_l\}_{l \geq 0}$ is bounded and decreasing.

- By Lemma 3, $\lim_{l \to \infty} b_l = 0$ w.p.1 with Assumption (2b), (2c), (2d), (2e).

- Then we show that for any $N \in \mathbb{N}^+$, $\lim_{l \to \infty} \left( \sup_{k:n \leq k \leq n+N} || \sum_{i=n}^{k} \alpha_i w_i || \right) = 0$.

    Define $M_n = \sum_{i=1}^{n} \alpha_i w_i$. Then $M_n = M_{n-1} + \alpha_n w_n$. As the set $\{\theta_i\}_{i=1}^{n_l}$ is generated i.i.d. with distribution $f_{m^{-1}(\hat{\eta}_l)}(\cdot)$ and $\hat{\eta}_l = \mathbb{E}_{m^{-1}(\hat{\eta}_l)}[\Gamma(\theta)]$,

$$\mathbb{E}[M_n | \sigma(M_1, \ldots, M_{n-1})]$$

$$= M_{n-1} + \mathbb{E}_{n^{-1}(\hat{\eta}_l)}[\frac{1}{n_l} \sum_{i=1}^{n_l} \Gamma(\theta_i) | M_{l-1}] - \hat{\eta}_l = M_{n-1}$$

    regardless of the value of $\hat{\eta}_l$. Therefore $\{M_n\}_{n \geq 0}$ is a martingale. Note that $w_i$ is independent on $w_j$ if $i \neq j$, as all $\theta$ are independently generated. Therefore $\mathbb{E}[w_i^\mathsf{T} w_j] = \mathbb{E}[w_i]^\mathsf{T} \mathbb{E}[w_j] = 0$.

$$\mathbb{E}[||M_n||^2]$$

$$= \mathbb{E}[M_n^\mathsf{T} M_n] = \mathbb{E}[(\sum_{i=1}^{n} \alpha_i w_i)^\mathsf{T} (\sum_{i=1}^{n} \alpha_i w_i)]$$

$$= \sum_{i=1}^{n} \alpha_i^2 \mathbb{E}[w_i^\mathsf{T} w_i] + \sum_{i=1}^{n} \sum_{j \neq i} \alpha_i \alpha_j \mathbb{E}[w_i^\mathsf{T} w_j]$$

$$= \sum_{i=1}^{n} \alpha_i^2 \mathbb{E}[w_i^\mathsf{T} w_i]$$

$$= \sum_{i=1}^{n} \frac{(1 - \alpha_i)^2 \lambda_i^2}{n_i} \text{Cov}_{m^{-1}(\hat{\eta}_i)}[\Gamma(\theta)].$$

As $\Gamma(\theta)$ is continuous and the domain $\Theta$ is compact, there exists $0 < C < \infty$ such that $\text{Cov}_{\boldsymbol{v}}[\Gamma(\theta)] \leq C$ for any $\boldsymbol{v} \in \mathcal{V}$. Therefore by Assumption (2b),

$$\mathbb{E}[||M_n||^2] \leq \sum_{i=1}^{n} C \frac{(1 - \alpha_i)^2 \lambda_i^2}{n_i} = O(\sum_{l=1}^{n} \frac{1}{l^{\beta + 2\lambda}}).$$

By Assumption (2b), $\beta + 2\lambda > 1$ and thus $\lim_{n \to \infty} \mathbb{E}[||M_n||^2] < \infty$. As $\{||M_n||^2\}$ increases monotonically, we know $\sup_n \mathbb{E}[||M_n||^2] = \lim_{n \to \infty} \mathbb{E}[||M_n||^2] < \infty$. Then by $L_2$ martingale convergence theorem, there exists $M_\infty$ such that $M_n \to M_\infty$ w.p.1 and $\mathbb{E}[||M_\infty||^2] < \infty$.

$$\sup_{\{k:n \leq k \leq n+N\}} || \sum_{i=n}^{k} \alpha_i w_i ||$$

$$= \sup_{\{k:n \leq k \leq n+N\}} || M_k - M_{n-1} || \leq 2 \sup_{k \geq n} || M_k ||.$$

Therefore

$$0 \leq \lim_{n \to \infty} \left( \sup_{\{k:n \leq k \leq n+N\}} || \sum_{i=n}^{k} \alpha_i w_i || \right)$$

$$\leq \lim_{n \to \infty} \left( 2 \sup_{k \geq n-1} || M_k || \right) = 0$$

    for any finite $N > 0$.

Since all conditions in Theorem 1 are satisfied, the limit set of sequence $\{\hat{\eta}_l\}_{l \geq 0}$ is a internally chain recurrent connected set for the flow induced by $\bar{L}(\eta) := \frac{\mathbb{E}_{\boldsymbol{v}}[G(\pi_\theta) S(\pi_\theta, \boldsymbol{v}, \rho) \Gamma(\theta)]}{\mathbb{E}_{\boldsymbol{v}}[G(\pi_\theta) S(\pi_\theta, \boldsymbol{v}, \rho)]} \Big|_{\boldsymbol{v} = m^{-1}(\eta)} - \eta$ w.p.1.

By (6), $\bar{L}(\eta) = \left( \frac{\partial}{\partial \boldsymbol{v}} \log L(\boldsymbol{v}; \rho) \right)^\mathsf{T} \Big|_{\boldsymbol{v} = m^{-1}(\eta)}$, which coincides with the right hand side of (8). $\square$

**Proof of Theorem 4.2**

Now we restate Theorem 4.2 and give a proof.

**Theorem 4.2.** *Let $\varphi : \mathcal{V} \to \mathbb{R}$ be any function such that $\frac{\partial}{\partial \boldsymbol{v}} \varphi(\boldsymbol{v}) = \tilde{L}(\boldsymbol{v}; \rho)$. Any equilibrium $\bar{\boldsymbol{v}}^* \in int(\mathcal{V})$ of (9) that is an isolated local maximum of $\varphi(\boldsymbol{v})$ is locally asympototically stable.*

*Proof.* The Lyapunov function we use is similar to that in [Joseph and Bhatnagar, 2016]:

$$V(\boldsymbol{v}) := \varphi(\bar{\boldsymbol{v}}^*) - \varphi(\boldsymbol{v}),$$

where $\bar{\boldsymbol{v}}^*$ is an isolated local maximum of $\varphi(\boldsymbol{v})$ and $\boldsymbol{v}$ is in some neighborhood of $\bar{\boldsymbol{v}}^*$ such that $\varphi(\bar{\boldsymbol{v}}^*) \geq \varphi(\boldsymbol{v})$, i.e. $V(\boldsymbol{v}) \geq 0$. By previous analysis, $\log \varphi(\boldsymbol{v})$ and $V(\boldsymbol{v})$ are continuous in $\boldsymbol{v}$. For the derivative:

$$\frac{dV(\boldsymbol{v})}{dt} = -\frac{\partial \boldsymbol{v}}{\partial t} \frac{\partial \varphi(\boldsymbol{v})}{\partial \boldsymbol{v}} = -\left(\tilde{L}(\boldsymbol{v}; \rho)\right)^{\mathsf{T}} (\text{Cov}[\Gamma(\theta)])^{-1} \tilde{L}(\boldsymbol{v}; \rho).$$

As $\text{Cov}_{\boldsymbol{v}}[\Gamma(\theta)]$ is positive definite for $\boldsymbol{v} \in int(\mathcal{V})$, $\left(\text{Cov}_{\boldsymbol{v}}[\Gamma(\theta)]\right)^{-1}$ is also positive definite. Therefore $\frac{\partial V(\boldsymbol{v})}{\partial t} \leq 0$ in a neighborhood of $\boldsymbol{v}^*$ and $\frac{\partial V(\boldsymbol{v})}{\partial t} = \boldsymbol{0}$ if and only if $\tilde{L}(\boldsymbol{v}; \rho) = \boldsymbol{0}$, which guarantees that $\boldsymbol{v}$ is a stationary point of (9). As $\bar{\boldsymbol{v}}^*$ is an isolated local maximum of $\varphi(\boldsymbol{v})$, it is the only stationary point in some neighborhood of $\bar{\boldsymbol{v}}^*$. Therefore $\frac{\partial V(\boldsymbol{v})}{\partial \boldsymbol{v}} = \boldsymbol{0}$ if and only if $\boldsymbol{v} = \bar{\boldsymbol{v}}^*$ (if $\boldsymbol{v}$ is in the neighborhood of $\boldsymbol{v}^*$) and $\bar{\boldsymbol{v}}^*$ is locally asymptotically stable. $\qquad\square$

In order to state the result we need to first introduce some definitions. By Assumption (2a), $Z := \left(\tilde{L}(\boldsymbol{v}; \rho)\right)^{\mathsf{T}} (\text{Cov}_{\boldsymbol{v}}[\Gamma(\theta)])^{-1}$ is a continuous vector field defined on $\mathcal{V} \subset \mathbb{R}^{d_v}$ with unique integral curves. The *flow* of $Z$ is the family of mappings $\{\Phi_t(\cdot)\}_{t \in \mathbb{R}}$ defined on $\mathcal{V}$ by $\frac{\partial \Phi_t(\boldsymbol{v})}{\partial t} = Z(\Phi_t(\boldsymbol{v}))$ such that $\Phi_0(\boldsymbol{v}) \equiv \boldsymbol{v}$ and $\Phi_t(\Phi_s(\boldsymbol{v})) \equiv \Phi_{t+s}(\boldsymbol{v})$ for any $\boldsymbol{v} \in \mathcal{V}$, $t, s \in \mathbb{R}$. $\boldsymbol{v} \in \mathcal{V}$ is an *equilibrium* if $\Phi_t(\boldsymbol{v}) = \boldsymbol{v}$ for all $t$. A set $\mathcal{V}' \subset \mathcal{V}$ is *positively invariant* under the flow $\Phi$ if for all $t \geq 0$, $\Phi_t(\mathcal{V}') = \mathcal{V}'$.

**Proof of Theorem 4.3**

**Theorem 4.3.** *If all equilibria of (9) are isolated, the sequence $\{\boldsymbol{v}_l\}_{l \geq 0}$ derived by Algorithm 1 converges toward an equilibrium of (9) as $l \to \infty$ with probability 1.*

*Proof.* Let $\varphi$ be defined in the same way as in Theorem 4.2. We first show that $\varphi$ is bounded over $\mathcal{V}$. By definition of $\tilde{L}(\boldsymbol{v}; \rho)$ in (6) of Section 4.2, $\tilde{L}(\boldsymbol{v}; \rho) = \frac{\mathbb{E}_{\boldsymbol{v}}[G(\pi_\theta) S(\pi_\theta, \boldsymbol{v}, \rho) \Gamma(\theta)]}{L(\boldsymbol{v}; \rho)} - m(\boldsymbol{v})$. Since $G$ has a positive lower bound (by Assumption (2e)) and $\mathbb{E}_{\boldsymbol{v}}[S(\pi_\theta, \boldsymbol{v}', \rho)] \geq \rho$ for any $\boldsymbol{v} \in \mathcal{V}$, $L(\boldsymbol{v}; \rho) \geq \inf_{\pi \in \Pi} G(\pi) \rho > 0$. Since $\Gamma$ is continuous over $\Theta$, $\Theta$ and $\mathcal{V}$ are compact (by Assumption (2d)), $\Gamma(\theta)$ and $m(\boldsymbol{v}) = \mathbb{E}_{\boldsymbol{v}}[\Gamma(\theta)]$ are both bounded. Since $G$ is also bounded (by Assumption (2e)), $\mathbb{E}_{\boldsymbol{v}}[G(\pi_\theta) S(\pi_\theta, \boldsymbol{v}, \rho) \Gamma(\theta)]$ is also bounded over $\mathcal{V}$ for any $\rho \in (0, 1)$. Therefore $\varphi$ is also bounded over $\mathcal{V}$.

Let $\Phi$ be a flow induced by (9) in Section 4.3 and $\Lambda$ be the set of all equilibria of (9). By definition, $\Lambda$ is positively invariant under $\Phi$. Define $V : \mathcal{V} \to \mathbb{R}^{\geq 0}$ as $V(\boldsymbol{v}) := \sup_{\boldsymbol{v}' \in \mathcal{V}} \varphi(\boldsymbol{v}') - \varphi(\boldsymbol{v})$. $\sup_{\boldsymbol{v}' \in \mathcal{V}} \varphi(\boldsymbol{v}') < \infty$ as $\varphi$ is shown to be bounded in $\mathcal{V}$. By definition of $\Lambda$ and the proof of Theorem 4.2, the mapping $t \mapsto V(\Phi_t(\boldsymbol{v}))$ is constant-valued for $\boldsymbol{v} \in \Lambda$ and strictly decreasing for $\boldsymbol{v} \in int(\mathcal{V}) \backslash \Lambda$. Since we also assume that (9) has only isolated equilibria and $\boldsymbol{v}$ is always in the interior of $\mathcal{V}$ (Assumption (2f)), $\{\boldsymbol{v}_l\}_{l \geq 0}$ converges to an equilibrium of (9) as $l \to \infty$ with probability 1 by Corollary 3.3 in [Benaim, 1996]. $\qquad\square$

## Experiment Details

**Environment map and the local sensing model** The robot's state space is $\mathcal{S} = \{(x, y, \zeta) | x_{min} \leq x \leq x_{max}, y_{min} \leq y \leq y_{max}, -\pi \leq \zeta < \pi\}$, which contains the agent's position and orientation in the global coordinate; the input space is 2-dimensional: $\mathcal{A} = \{(v, \omega) | |v| \leq v_{max}, |\omega| \leq \omega_{max}\}$, which are linear and angular speed respectively. We assume that the robot can control $v$ and $\omega$ directly.

(a)            (b)

Figure 1: (1a) Map of the car navigation example. There are one obstacle region (big grey rectangle), one goal region (small blue rectangle) and 10 randomly selected initial states (red circles pointing to the forward direction). Dotted lines are added to show $x$ and $y$ axes. (1b) Illustrations of local features in the agent's local coordinate at one of the initial states, with $n_s = 5$. Obstacle nodes, goal nodes and free nodes are labeled by black crosses, yellow plus signs and green triangles respectively. The goal direction (shown as the black arrow) is also included in local features.

There is a goal region $\mathcal{G}$ and a non-overlapping bad region $\mathcal{B}$ such that $\mathcal{G}, \mathcal{B} \subset [x_{min}, x_{max}] \times [y_{min}, y_{max}]$. The map is shown in Figure 1a.

Since the robot has only local sensors, we use the following local sensing model instead of assuming the knowledge of the true state variables $x, y$ and $\zeta$. For a given positive integer parameter $n_s$, we design a radial grid as $n_s$ circles in the agent's local coordinate. The difference between the diameters of adjacent circles is $v_{max}\Delta t$, where $\Delta t$ is the sampling time. There are $\lceil 2\pi/\omega_{max} \rceil$ uniformly distributed nodes on each circle and the robot can measure the label for each node. A node is labeled 1 if it belongs to $\mathcal{G}$; -1 if it belongs to $\mathcal{B}$ and 0 otherwise. We also assume that the robot can sense the direction of the center of $\mathcal{G}$ in its local coordinate without knowing the distance, so there are a total of $(2 + n_s\lceil 2\pi/\omega_{max} \rceil)$ local features in total. The local features are illustrated in Figure 1b. In our experiment, $\omega_{max} = \frac{\pi}{6}$, $n_s = 5$, so there are 62 local features as the inputs to the policy network. Note that the local sensor outputs are all discrete and only 2 features are continuous (the goal direction in the agent's local coordinate), so the problem is much simpler than a general continuous RL problem with the same number of continuous inputs.

**Algorithm parameters** In all experiments, we set $F_{\mathcal{V}}$ as a class of multivariate Gaussian distributions with diagonal covariance matrices. The parameter space $\Theta$ contains all the parameters of the policy network. The policy space $\Pi_\Theta$ is a set of deterministic stationary policies. Therefore the CCE trains a single neural network which takes states as inputs and output a single action. The two baseline algorithms TRPO Schulman et al. [2015] and CPO Achiam et al. [2017] take Gaussian policies, which takes states as inputs and outputs the mean and variance of the action distribution.

The policy networks for all experiments have two hidden layers of sizes $(30, 30)$. The activation function for hidden layers is ReLU and that for the output layer is tanh. The length of sample trajectories are all 30. The same set of parameters are applied for all experiments. For CCE, we sample 40 different policies in each iteration. Each sampled policy is evaluated using 10 sample trajectories. The hyperparameter for selecting elite examples is $\rho = 0.2$. For both CPO and TRPO, the batch size is 6000, discount factor is 0.999, and the step size for trust region is 0.01. All the other parameters are used as default in the source code in rllab Duan et al. [2016].

**The axes in the learning curve (Figure 1 in the paper)** The x-axes in Figure 1 show the total number of sample trajectories for CCE or the total number of equivalent sample trajectories for TRPO and CPO. Assume that in each iteration of the CCE algorithm, we sample 40 policies (i.e., $n_l = 40$) and simulate 10 sample trajectories for each policy (Step 5 of Algorithm 1), then the total number of sample trajectories is 400 per iteration. If trajectory length is 30 and we sample 6000 new transitions in each iteration for CPO and TRPO, the number of equivalent sample trajectories is 200 per iteration. As we set the same trajectory length for all methods, the numbers of sample trajectories for all methods are comparable with each other.

The y-axes in Figure 1 show the *average* objective and constraint values of the learned policy. For CCE, the average values are computed with all rollout trajectories that are simulated with *all* the

policies sampled at the current iteration. Since we take the same number of rollout trajectories for each sample policy, the average value can be interpreted as the average performance of all sample policies at the current iteration. For CPO and TRPO, we simulate the current policy from exactly the same set of initial states and compute the average objective and constraint values for all trajectories. As a result, the comparison of different methods in Figure 1 is fair.