[Reviews · NeurIPS 2018]

Reviewer 1



This paper studies constrained optimal control, where the goal is to produce a policy that maximizes an objective function subject to a constraint. The authors provide great motivation for this setting, explaining why the constraint cannot simply be included as a large negative reward. They detail challenges in solving this problem, especially if the initial policy does not satisfy the constraint. They also note a clever extension of their method, where they use the constraint to define the objective, by setting the constraint to indicate whether the task is solved. Their algorithm builds upon CEM: at each iteration, if there are no feasible policies, they maximize the constraint function for the policies with the largest objective; otherwise, they maximize the objective function for feasible policies. Overall, the paper is technically rigorous, while also providing high-level intuition to explain sections of dense math. Perhaps the largest weakness of the paper is that experiments are limited to a single task. That said, they compare against two reasonable baselines (CPO, and including the constraint as a negative reward). While the formal definition of the constrained objective in L149 - L155 is appreciated, it might be made a bit more clear by avoiding some notation. For example, instead of defining a new function I(x,y) (L151), you could simply use \delta(x >= y), stating that \delta is the Dirac delta function. A visualization of Eq 1 might be helpful to provide intuition. Minutia * Don't start sentences with citation. For example, (L79): "[32] proposed ..." * Stylistically, it's a bit odd to use a *lower* bound on the *cost* as a constraint. Usually, "costs" are minimized, so we'd want an upper bound. * Second part of Def 4.2 (L148) is redundant. If Pr(X >= \gamma) <= \rho, then it's vacuously true that Pr(X <= \gamma) >= 1 - \rho. Also, one of these two terms should be a strict inequality. * Include the definition of S (L154.5) on its own line (or use \mbox). * Label the X and Y axes of plots. This paper makes an important step towards safe RL. While this paper builds upon much previous work, it clearly documents and discusses comparisons to previous work. While the results are principally theoretical, I believe it will inspire both more theoretical work and practical applications.

Reviewer 2



This paper deals with a safe reinforcement learning problem, where constraints are defined as an expected cost over finite-length trajectories, the goal being to maximize an expected gain (also over finite-length trajectories, so more general than cumulative rewards), while the constraint is greater than a given lower bound. To do so, the authors extend the cross-entropy method applied to classic RL. If sampled policies are feasible (satisfy the constraints), they are sorted according to their gain (as in classic cross-entropy). If not, how much the constraints are violated is also taken into account (by selecting the closer to feasible ones). The proposed contribution is supported by a theoretical analysis and experimental results. Overall, the paper is well written, deals with an important subject, and provides interesting theoretical and experimental results. However, it could be improved with some clarifications, regarding the approach, as well as theoretical and empirical results. Also, the scalability of the proposed approach could be discussed (a possible work about this is “Evolution strategies as a scalable alternative to reinforcement learning”). Many (usually small) changes could help clarify the proposed approach, for example: * It would be better to say already in l.110 that J>0 (it is said after, but can be unnoticed, and Eq.1 makes no sense with negative J). * In Alg.1, what are N_l (n_l?), lambda_l? Also, in "ensure" part, there should be NEF and Pi * sec. 4.2 could me made clearer, notably by clarifying how one goes from distributions g to the etas, and especially how lines 11 and 12 in Alg.1 are derived (it is not that clear, even with the appendix). It could maybe also help to instantiate it (say in the appendix), for example for the Gaussian with diagonal matrix case considered practically. * l.230, "by definition of eta", this is really not clear. The definition is eta(t) = m(v(t)), but it is never clearly stated in the paper (can be infered from alg.1, or line 200) * In the main text (eg l.115), a policy is said to be feasible if H_Z(pi)>d. In Fig. 1, feasible policies are below the dashed lines, that is H_Z(pi)

Reviewer 3



In this paper, the authors studied a safe RL problem in which the constraints are defined over the expected cost over finite length trjectories. Based on the cross-entropy method, the method tracks the reward performance and constratint cost satisfaction and update accordingly the objective function. They further show that the asymptotic behavior of the algorithm can be described by an ordinary differential equation, and give sufficient conditions on the properties of this differential equation to guarantee the convergence of the proposed algorithm. Simulation shows that this method produces compelling results when compared with CPO, a constrained version of the TRPO algorithm. Furthermore, this algorithm can also effectively solves problem without assumptions on the feasibility of initial policies, even with non-Markovian objective functions and constraint functions. In general, I think this paper studies a very interesting and practical problem of constrained RL, which also has applications to multi-objective RL. Rather than using gradient-based policy optimization approaches, the authors turn to the evolutionary CE method, where a bunch of policy parameters are sampled from some prior distribution at each iteration, and the best portions of the samples are used to update the distribution of the policy parameter. Specifically, the policy parameter is assumed to have a NEF distribution. Since the RL problem is constrained, the loss function is of a hybrid form. If the (1−\rho)-quantile of constraint for the current policy distribution is less than the constraint threshold, we select policies by their constraint values in order to increase the probability of drawing feasible policies. On the other hand, if the proportion of feasible policies is higher than \rho, we only select feasible policies with large objective function values. In general, I found the CE procedure intuitive, and the description clearly written. Furthermore, the authors analyzed the CE algorithm using standard asymptotic analysis of stochastic approximation and ODE, and show that the constrained CE algorithm converges to a local maximum (and feasible) solution almost surely. Although I haven't had a chance to go through the proof details in the appendix, the analysis looks rigorous to me. Then they show that the constrained CE algorithm out-performs CPO in a simple robot navigation problem especially when the cost and constraint functions are non-Markovian, which is very interesting. However, I have the following technical questions/concerns: 1) Unlike Lagrangian approaches, CPO is a policy gradient method that is derived to guarantee feasibility at every update iteration. Although it's true that in function approximations, without engineering tricks such a guarantee in general might fail to hold, this algorithm is still developed based on principled arguments on constraint satisfaction versus policy improvement. But to the best of my knowledge, for evolutionary methods such as CE, while it can work surprisingly well on some examples, they have the disadvantage of updating parameter values stochastically, without following any explicit gradient that carries information about what changes to parameter values are likely to improve performance. In this case, I suspect the constrained CE can guarantee constraint satisfaction at every update. 2) Since convergence of the proposed algorithm is in the asymptotic sense, why don't we use a Lagrangian approach to solve the constraint problem (i.e., for fixed \lambda, we solve for the unconstrained problem using CE, and update \lambda in a slower time-scale with rewards/constraint cost sampled from the black box)? In this case, we can directly re-use existing algorithms and analysis of unconstrained CE. How does this approach numerically compare with the proposed constrained algorithm? 3) While I appreciate the positive results reported in this paper, (especially the non-Markovian ones) since CE is an evolutionary algorithm, testing only on one domain might not be convincing enough to conclude that constrained CE will always out-perform CPO. Have you compared the proposed method with CPO on the Mujoco benchmarks reported in the CPO paper? 4) Can we extend the constrained CE method to handle infinite horizon problems?